# Properties of fermionic systems
# with the path-integral ground state method

Sebastian Ujevic[1], Vinicius Zampronio[2*], Bruno R. de Abreu[3] and Silvio A. Vitiello[4]

**1** Departamento de Ciências Exatas-EEIMVR,
Universidade Federal Fluminense, 27255-125 Volta Redonda, RJ, Brazil
**2** Institute for Theoretical Physics, Utrecht University, 3584CS Utrecht, Netherlands
**3** National Center for Supercomputing Applications,
University of Illinois at Urbana-Champaign, Urbana, IL 61801, USA
**4** Instituto de Física Gleb Wataghin, University of Campinas - UNICAMP,
13083-859 Campinas - SP, Brazil

★ v.zamproniopedroso@uu.nl

## Abstract

We investigate strongly correlated many-body systems composed of bosons and fermions with a fully quantum treatment using the path-integral ground state method, PIGS. To account for the Fermi-Dirac statistics, we implement the fixed-node approximation into PIGS, which we then call FN-PIGS. In great detail, we discuss the pair density matrices we use to construct the full density operator in coordinate representation, a vital ingredient of the method. We consider the harmonic oscillator as a proof-of-concept and, as a platform representing quantum many-body systems, we explore helium atoms. Pure $^4$He systems demonstrate most of the features of the method. Complementarily, for pure $^3$He, the fixed-node approximation resolves the ubiquitous sign problem stemming from anti-symmetric wave functions. Finally, we investigate $^3$He-$^4$He mixtures, demonstrating the method's robustness. One of the main features of FN-PIGS is its ability to estimate any property at temperature $T = 0$ without any additional bias apart from the FN approximation; biases from long simulations are also excluded. In particular, we calculate the correlation function of pairs of equal and opposite spins and precise values of the $^3$He kinetic energy in the mixture.

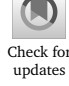
# 1  Introduction

Strongly interacting fermions are a building block of matter in scales ranging from nuclear physics to neutron stars [1–4]. Over the past several decades, efforts to unravel theories to describe these systems have raised several issues that attest to how challenging it can be to approach them comprehensively [5].

Mixtures of bosons and fermions are an active field of research with scientific interest renewed by the recent observation of quantum gas degeneracies [6–8]. A prominent example is helium systems, which have, for a long time now, been the bedrock of strongly interacting many-body physics. Despite that, they exhibit properties and features that are not entirely understood yet, such as the kinetic energy of isotopic $^3$He-$^4$He mixtures [9, 10]. One of the goals of this work is to investigate the kinetic energy of mixtures of $^3$He in the normal phase and $^4$He in the superfluid one, with both species receiving a fully quantum many-body treatment concerning the statistics they obey. For this purpose, we have revisited a quantum Monte Carlo (QMC) method at zero temperature [11], demonstrating its accuracy. It allows the quantum many-body treatment to be subjected exclusively to an approximation of the nodal structure of the wave function for the fermionic species. Moreover, $^3$He-$^4$He mixtures offer several of

the complexities needed for the method to display its capabilities within a well-known system.

Investigations of the $^3$He-$^4$He mixtures, where both Bose-Einstein and Fermi-Dirac statistics are involved, have a long history [12–14]. They are challenging systems for experimental and theoretical studies near zero temperatures. In experiments, the quest for a doubly superfluid Bose-Fermi of the mixture continues [15], despite extraordinary technical difficulties to reach its predicted critical temperature of about 45 $\mu$K [16]. Another challenge regarding $^3$He atoms occurs in experiments with neutrons, where an absorption cross-section of about $10^4$ barns [17] makes observations of the intensity of the scattered neutrons extremely difficult. On the theoretical side, a qualitative understanding of many characteristics of helium mixtures involves a delicate balance between several effects, including the local density, the Heisenberg uncertainty principle, an attraction between $^3$He atoms due to magnetic interaction caused by nuclear magnetic moments, the Pauli exclusion principle, and atomic binding energies [18]. Additionally, due to the strongly correlated character of helium atoms, attempts to approach this problem quantitatively require non-perturbative theoretical treatments.

Quantum Monte Carlo methods are incredibly convenient for clarifying a myriad of properties of quantum many-body systems in stationary states in an exact form, providing results for bosonic systems subjected solely to statistical uncertainties. In continuous space, systems at finite temperature can be investigated with the path-integral Monte Carlo (PIMC) method [19] or with its variant, the worm algorithm [20]. PIMC calculations have successfully deepened our understanding of the $^3$He-$^4$He mixtures [10, 21, 22]. At zero temperature, the Diffusion Monte Carlo (DMC) method can estimate energies without biases [23, 24]. Nevertheless, properties that do not commute with the Hamiltonian need to be extrapolated using variational results, a process that may not be unbiased [25]. The DMC method has also been used to study $^3$He-$^4$He mixtures [26].

Systems made of particles that obey Fermi-Dirac statistics offer an additional challenge to QMC methods that stems from sampling the relevant probability distributions. Quantities associated with these distributions are not always positive-definite due to the Pauli exclusion principle, *i.e.*, the antisymmetric character of the wave functions that describe these systems. This complication is at the heart of the notorious sign problem that permeates fermionic systems [5]. For a review concerning this issue and attempts to overcome it, see [27]. PIMC simulations of mixtures are performed by treating $^3$He atoms as boltzmannons or by applying the restricted path integral Monte Carlo as a way to circumvent the sign problem [28].

The simulation method we deploy originated from a generalization of the shadow wave function idea [29] and was named the *variational path integral* (VPI) method by its creator [19]. Later on, it was renamed [30, 31] to *path-integral ground state method* (PIGS).

In recent studies of bosonic systems, PIGS has often been the optimal choice among ground state QMC methods because it does not rely on importance sampling. PIGS calculations of two-dimensional dipolar systems resulted in the expected supersolid stripe phase without adding one-body localization factors to the trial wave function [32]. More surprisingly, the ground state of liquid helium can be projected from a constant function [33]. Another powerful feature of the PIGS method is the straightforward implementation of estimators that give unbiased estimates of local and non-local properties. Two very interesting non-local estimates of this sort are the one-body density matrix (OBDM) [33] and the entanglement entropy [34, 35]. In particular, the dynamics of one-dimensional liquid $^4$He were studied from PIGS estimates of the dynamical structure factor [36].

The PIGS method applies convolutions of a given approximation of the density matrix to a trial function to filter the system's ground state. The filtering process for bosonic systems is exact if the projection is long enough in imaginary time. In this work, we review in depth the PIGS method, including its extension to treat fermions using the fixed-node approximation. With a different focus, an excellent review of the method was put forward by Yan and

Blume [37].

PIGS estimates any property without biases, regardless of whether or not the property commutes with the system's Hamiltonian. In our approach for fermionic systems, all properties are estimated within the fixed-node approximation, where only a single given nodal region is considered, avoiding changes of sign in the sampled probability. The fixed-node approximation is straightforward to implement within PIGS. FN-PIGS is a state-of-the-art method for treating fermions, paving the way for further studies of local, non-local, and dynamical properties free from extrapolations.

In the following sections, we present details of the PIGS method and results for a proof-of-concept system. We then show results for pure $^4$He systems before reaching the primary purpose of our work, which is the investigation of fermionic systems. We investigate systems composed of $^3$He atoms and the much more complex case of the $^3$He-$^4$He mixture.

PIGS and PIMC methods are similar. Many essential aspects of the method can be found in the magnificent review by David Ceperley about path integrals [19]. Here, we give technical details and particularities of the numerical treatment involved in PIGS. We also provide a detailed description of how estimators are constructed, particularly for the radial distribution function.

Finally, we mention that we have limited the study of $^3$He-$^4$He mixtures to some of their basic properties, leaving aside other important and interesting characteristics of these quantum liquids. For instance, we have yet to attempt to apply the worm algorithm to evaluate the one-body density matrix and the condensate fraction of the $^4$He isotope. With our work as a stepping stone, we expect these properties to be investigated in the future.

## 2 The PIGS method

The formal solution of Schrödinger's equation is the time evolution operator $\hat{K}(t) = e^{-it\hat{H}}$, where $t$ is the time and $\hat{H}$ is the Hamiltonian of the quantum system. This operator is intrinsically connected to the density operator in statistical mechanics, $\hat{\rho}(\tau) = e^{-\tau\hat{H}}$, where $\tau = 1/k_B T$ is the inverse temperature, $k_B$ is the Boltzmann constant, and $T$ is the temperature. One can transform $\hat{\rho}$ into $\hat{K}$ by applying the so-called Wick rotation $\tau \rightarrow it$. Therefore, $\tau$ is also commonly referred to as the *imaginary time*. This transformation leads to several approaches that explore a quantum-to-classical mapping to calculate statistical properties and expectation values in quantum systems without dealing with integration measures that are not positive-definite or even real, such as the renowned PIMC method [19].

In the PIGS method the density matrix $\hat{\rho}(\tau) = \exp(-\tau\hat{H})$ projects a trial state $|\Psi_T\rangle$ onto a quantum state $|\Psi_\tau\rangle$

$$|\Psi_\tau\rangle = e^{-\tau\hat{H}} |\Psi_T\rangle \,, \tag{1}$$

where $\tau$ is a real number. We assume that all quantum states are not normalized. If $|\Psi_T\rangle$ is non-orthogonal to the ground state $|\Psi_0\rangle$, the projection is guaranteed to exponentially approach $|\Psi_0\rangle$ as $\tau$ increases. This can be easily verified expanding $|\Psi_T\rangle$ into eigenstates of the Hamiltonian,

$$|\Psi_\tau\rangle = e^{-\tau E_0} c_0 |\Psi_0\rangle + e^{-\tau E_1} c_1 |\Psi_1\rangle + \dots \,, \tag{2}$$

where $E_i < E_{i+1}$ are eigenvalues of $\hat{H}$ and $c_i$ are expansion coefficients. For a large enough $\tau$, the term with the smallest decay rate, namely the ground state, entirely dominates the expansion.

In practice, to carry on computer simulations, these projections are performed in coordinate space representation, where the density operator has the matrix elements

$$\rho(R,R',\tau) = \langle R|\hat{\rho}(\tau)|R'\rangle, \tag{3}$$

where $R \equiv \{\mathbf{r}_i \mid i = 1,\dots N\}$ is the set of coordinates $\mathbf{r}_i$ of all $N$ particles in the system, which can be of any spatial dimension. The PIGS method takes advantage of the convolution property of the density matrix, also called imaginary time composition, which reads

$$\rho(2\tau) = \rho(\tau)\rho(\tau), \tag{4}$$

and which becomes an explicit convolution operation in coordinate space:

$$\rho(R_i,R_j,2\tau) = \int dR_1 \rho(R_i,R_1,\tau)\rho(R_1,R_j,\tau). \tag{5}$$

The last equation is an *exact* expression of the density matrix for any imaginary time $\tau$.

Apart from normalization constants, the ground state expectation value of any quantity $O$ associated with the operator $\hat{O}$,

$$O \propto \langle \Psi_0|\hat{O}|\Psi_0\rangle, \tag{6}$$

can be estimated in the following way. Consider initially $O(\tau)$ given by

$$O(\tau) \propto \langle \Psi_\tau|\hat{O}|\Psi_\tau\rangle = \langle \Psi_T|e^{-\tau\hat{H}}\hat{O}e^{-\tau\hat{H}}|\Psi_T\rangle. \tag{7}$$

We can insert completeness relations $\hat{1} = \int dR_i|R_i\rangle\langle R_i|$ to the left side of $|\Psi_T\rangle$ and to the right side of $\langle\Psi_T|$ such that

$$O(\tau) \propto \int dR_{-M}dR_{M+1}\Psi_T^*(R_{-M})\langle R_{-M}|e^{-\tau\hat{H}}\hat{O}e^{-\tau\hat{H}}|R_{M+1}\rangle\Psi_T(R_{M+1}), \tag{8}$$

with $\Psi_T^*(R_{-M}) = \langle\Psi_T|R_{-M}\rangle$ and $\Psi_T(R_{M+1}) = \langle R_{M+1}|\Psi_T\rangle$. Next, we use imaginary time composition to slice the density operator on the right side of $\hat{O}$ into $M$ operators $e^{-\delta\tau H}$, with $\delta\tau = \tau/M$, such that

$$O(\tau) \propto \int dR_{-M}dR_{M+1}\Psi_T^*(R_{-M})\langle R_{-M}|e^{-\tau\hat{H}}\hat{O}e^{-\delta\tau\hat{H}}e^{-\delta\tau\hat{H}}\cdots e^{-\delta\tau\hat{H}}|R_{M+1}\rangle\Psi_T(R_{M+1}). \tag{9}$$

Insert another completeness relation to the left of the right-most short-imaginary time operator,

$$\begin{aligned} O(\tau) \propto \int dR_{-M}dR_M dR_{M+1} \\ \times \Psi_T^*(R_{-M})\langle R_{-M}|e^{-\tau\hat{H}}\hat{O}e^{-\delta\tau\hat{H}}e^{-\delta\tau\hat{H}}...\, e^{-\delta\tau\hat{H}}|R_M\rangle\langle R_M|e^{-\delta\tau\hat{H}}|R_{M+1}\rangle\Psi_T(R_{M+1}), \end{aligned} \tag{10}$$

or in terms of $\rho(R_M,R_{M+1};\delta\tau) = \langle R_M|e^{-\delta\tau\hat{H}}|R_{M+1}\rangle$,

$$\begin{aligned} O(\tau) \propto \int dR_{-M}dR_M dR_{M+1} \\ \times \Psi_T^*(R_{-M})\langle R_{-M}|e^{-\tau\hat{H}}\hat{O}e^{-\delta\tau\hat{H}}e^{-\delta\tau\hat{H}}...\, e^{-\delta\tau\hat{H}}|R_M\rangle\rho(R_M,R_{M+1};\delta\tau)\Psi_T(R_{M+1}). \end{aligned} \tag{11}$$

We are left with $(M-1)$ short-imaginary time operators. By repeating this procedure until all the slices are covered, we have

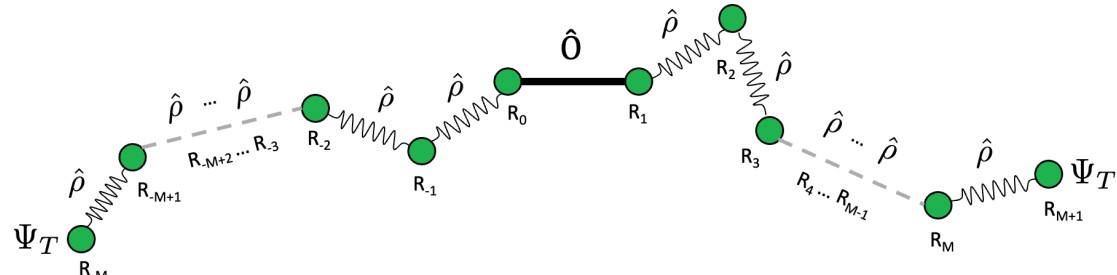

Figure 1: Visual representation of Eq. (13) where the configurations $R_i$ are represented by beads connected through density matrix elements $\rho(R_i, R_{i+1}; \delta\tau)$, represented by springs. The central beads $R_0$ and $R_1$ are connected by $\langle R_0|\hat{O}|R_1\rangle$, which is represented by a solid bar. The total number of beads in the chain is $2M+2$, where $M$ is the number of imaginary time compositions used to construct $e^{-\tau H}$. The endpoints $R_{-M}$ and $R_{M+1}$ also carry weights associated with the trial wave function $\Psi_T$.

$$O(\tau) \propto \int dR_{-M} \left(\prod_{i=1}^{M+1} dR_i\right) \Psi_T^*(R_{-M})\langle R_{-M}|e^{-\tau H}\hat{O}|R_1\rangle\rho(R_1, R_2; \delta\tau)\rho(R_2, R_3; \delta\tau)$$
$$\cdots\rho(R_M, R_{M+1}; \delta\tau)\Psi_T(R_{M+1}), \tag{12}$$

where we can see the formation of a "chain" of density matrix elements $\rho(R_i, R_{i+1}; \delta\tau)$ with endpoints $R_1$ and $R_{M+1}$, that are then integrated over the entire space. The same procedure can be repeated to slice $e^{-\tau H}$ on the left side of $\hat{O}$ into $M$ $e^{-\delta\tau H}$ operators and then insert intermediate configurations from $R_{-M+1}, ..., R_0$, which yields

$$O(\tau) \propto \int \left(\prod_{i=-M}^{M+1} dR_i\right) \Psi_T^*(R_{-M})\rho(R_{-M}, R_{-M+1}; \delta\tau)\cdots\rho(R_{-1}, R_0; \delta\tau)\langle R_0|\hat{O}|R_1\rangle\rho(R_1, R_2; \delta\tau)$$
$$\cdots\rho(R_M, R_{M+1}; \delta\tau)\Psi_T(R_{M+1}). \tag{13}$$

This equation has a visual representation that is shown in Fig. 1. Each of the $2M+2$ configurations $R_i$, represented as a bead, is linked to $R_{i+1}$ through the matrix element $\rho(R_i, R_{i+1}; \delta\tau)$, which is represented by a spring. The exception is the central configurations $R_0$ and $R_1$, connected by the operator $\hat{O}$, which is represented by a horizontal bar. The endpoint configurations $R_{-M}$ and $R_{M+1}$ carry the trial wave function $\Psi_T$.

## 2.1 Non-local operators

The probabilistic nature of $\rho$ and $\Psi_T$ offers the possibility of using Monte Carlo techniques to sample the configurations $R_i$, with $i \neq 0, 1$. To sample the central configurations $R_0$ and $R_1$, we are left with a choice to use any suitable probability distribution $p(R_0, R_1)$, since Eq. (13) can also be written as

$$O(\tau) \propto \int \left(\prod_{i=-M}^{M+1} dR_i\right) \Psi_T^*(R_{-M})\rho(R_{-M}, R_{-M+1}; \delta\tau)\dots\rho(R_{-1}, R_0; \delta\tau)O(R_0, R_1)\rho(R_1, R_2; \delta\tau)$$
$$\dots\rho(R_M, R_{M+1}; \delta\tau)\Psi_T(R_{M+1}), \tag{14}$$

where

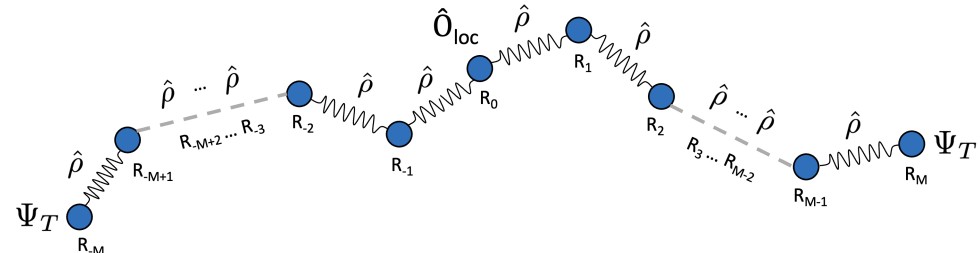

Figure 2: Visual representation of the PIGS method for the particular case when $\hat{O}$ is local in configuration space. The total number of beads is $2M+1$, and $O$ is calculated in the central bead $R_0$. Density matrix elements connect all internal beads. Therefore, the same algorithm can be used to sample all configurations $R_i$.

$$O(R_0, R_1) = \frac{\langle R_0|p(R_0, R_1)\hat{O}|R_1\rangle}{p(R_0, R_1)}, \tag{15}$$

such that $R_0$ and $R_1$ are amenable to importance sampling. The most convenient choice is to take $p(R_0, R_1) = \rho(R_0, R_1; \delta\tau)$, such that $R_0$ and $R_1$ are also connected by a spring, and the same algorithm can be used to sample the configurations of all the beads $R_{-M}, ..., R_{M+1}$ in the chain. Despite its convenience, this may not be the best choice depending on the nature of $\hat{O}$, and a more appropriate $p$ can be selected case-by-case. Finally, $O(\tau)$ can be estimated by sampling the probability distribution

$$\mathcal{P}(R_{-M}, \ldots, R_0, R_1, \ldots R_{M+1}) \propto \Psi_T^*(R_{-M})\left(\prod_{i=-M}^{-1} \rho(R_i, R_{i+1}; \delta\tau)\right)p(R_0, R_1)$$
$$\times \left(\prod_{i=1}^{M} \rho(R_i, R_{i+1}; \delta\tau)\right)\Psi_T(R_{M+1}). \tag{16}$$

## 2.2 Local operators

A particular relevant case of the PIGS method happens when the operator $\hat{O}$ is local in coordinate representation, such that:

$$\langle R_0|\hat{O}|R_1\rangle = O(R_1)\delta(R_0 - R_1), \tag{17}$$

forcing the configurations $R_1$ and $R_0$ to be the same. The integration in one of these coordinates can therefore be analytically performed, effectively collapsing the two central beads into a single one. This results in a chain of $2M+1$ beads, all connected by density operators $\rho(R_i, R_{i+1}; \delta\tau)$, as shown in Fig. 2.

In this particular case, one does not need to worry about choosing $p$, and estimates for $O(\tau)$ can be obtained by sampling the probability distribution

$$\mathcal{P}_{\text{loc}}(R_{-M}, \ldots, R_0, \ldots, R_M) \propto \Psi_T^*(R_{-M})\left(\prod_{i=-M}^{M-1} \rho(R_i, R_{i+1}; \delta\tau)\right)\Psi_T(R_M). \tag{18}$$

Ground state converged configurations must be used when evaluating the local quantities $O(R_0, R_1)$. In particular, it is imperative that the configurations $(R_0, R_1)$, points at which $\hat{O}$ is effectively evaluated, satisfy this ground state convergence condition. Another interesting

point here, which contrasts this method to others that share the same imaginary time composition approach enclosed in Eq. (7) through Eq. (13), is that, despite having a total of $(2M+2)$ beads (or intermediary configuration), we are slicing the imaginary time component into $M$ chunks. Configurations from $\Psi_T$ (endpoints) can be sampled via the Metropolis-Hastings algorithm. Those of the remaining beads can be efficiently sampled by a multi-level Metropolis algorithm (see Sec. 5).

In summary, the PIGS method estimates ground state properties without introducing any biases. This calculation is done through the application of a local operator $\hat{O}$ in converged configurations of the ground state, which are obtained by projecting a trial quantum state represented by the wave function $\Psi_T$ at both extremities of a linear polymer using the density matrix of the system. Estimates can be refined by adding more beads and/or applying the estimator $O$ to all beads with converged configurations when $[\hat{O}, \hat{\rho}] = 0$.

To fully account for quantum fluctuations, the probability distribution $\mathcal{P}$ must include Bose or Fermi statistics,

$$\Psi(R, \tau) = \frac{1}{N!} \sum_P \int dR' (\pm 1)^P \rho(R, \hat{P}R', \tau) \Psi_T(R'), \qquad (19)$$

where $\hat{P}$ is the operator that permutes $P$ particles of the system. In the PIGS method, the appropriate quantum statistics of the system can be implemented through the trial state $\Psi_T$. In this case, then

$$(\pm 1)^P \rho(R, \hat{P}R', \delta\tau) \Psi_T(R') = \rho(R, \hat{P}R', \delta\tau) \Psi_T(\hat{P}R') \rightarrow \rho(R, R', \delta\tau) \Psi_T(R'), \qquad (20)$$

and all terms in the sum of Eq. (19) are the same under the integration after relabelling particle indexes. Thus, it is possible to consider the density operator elements $\rho(R, R'; \delta\tau)$ for distinguishable particles [11,19]. For fermionic systems, $\Psi_T$ must be anti-symmetric under particle exchange. For this reason, $\mathcal{P}$ may change sign, which is the source of the sign problem in the PIGS method. The fixed-node approximation, which will be discussed in the next section, can be used to circumvent this problem.

## 3 Fixed-node approximation

The fixed-node approximation is a common way to address the sign problem in QMC methods. It was envisioned by Anderson [38], and its first application to the investigation of the electron gas ground state attained results that became an essential ingredient for the density functional theory [39,40]. In a somewhat more intricate fashion, similar ideas were employed to restrict path integrals from sign changes [28]. In the PIGS method, the fixed-node approximation can be implemented straightforwardly.

In fermionic systems, the nodes of the ground state wave function $\Psi_0$ carve up the configuration space into nodal regions. Ceperley demonstrated that two distinct nodal regions of the ground state differ only by the permutations of particle indexes, *i.e.* by the sign of $\Psi_0$ [28]. Consequently, properties associated with a given nodal region $\Omega$ are identical to those of other nodal regions. Thus, one can solve Schrödinger's equation inside one of the $\Omega$ regions and still obtain expectation values pertinent to the entire system.

In practice, the nodal hypersurfaces of a given *ansatz* $\Psi_T$ for the ground state are converted into infinite potential walls, such that all configurations $R_i$, with $i = \{-M, \dots, 0, 1, \dots, M+1\}$, are forced to remain inside the initial, arbitrary nodal region. In general, the nodal regions of $\Psi_0$ are unknown and are likely different than those of $\Psi_T$. Nevertheless, it has been shown

that, by using a single $\Omega$, one obtains the lowest energy estimate of the system, consistent with the nodal structure of the trial state [23, 28].

In a typical simulation, Monte Carlo moves are rejected if a configuration of the trial wave function or that of any bead crosses the nodal hyper-surface. A test to determine if the surface has been crossed can be established by verifying if the sign of $\Psi_T(R_i)$ has changed after the trial displacement. Cross-recross errors happen when a configuration $R_i$ leaves a nodal region with a particular sign and ends in another region with the same sign as the initial one. To avoid this error, we use sufficiently small values of the imaginary time step $\delta\tau$, which results in acceptance ratios greater than 90%.

To account for the infinite potential walls at the nodal hyper-surface, the density matrix $\rho(R,R';\delta\tau)$ must smoothly vanish at these boundaries. This is accomplished by implementing the image action approximation [41]. For instance, in a short-time $\delta\tau$ approximation, one can multiply $\rho(R,R';\delta\tau)$ by $e^{S_I}$, where

$$S_I = \ln\left\{1 - \exp\left[-\frac{d(R)d(R')}{\lambda\delta\tau}\right]\right\}, \tag{21}$$

$\lambda = \hbar^2/2m$, and $d(R)$ is the distance between configuration $R$ and the closest nodal region. The use of this approximation is strongly recommended. Without that, there would be a non-vanishing probability of finding beads in the proximity of the nodes. An exact analytical expression for $d(R)$ is difficult to obtain, but one can rely on the Newton-Raphson method, which gives the approximation

$$d(R) \approx \frac{|\Psi_T(R)|}{|\nabla\Psi_T(R)|}. \tag{22}$$

## 4 The density matrix

A central ingredient of the PIGS method is the representation of the density operator of the system $e^{-\tau\hat{H}}$ in coordinate space or, in other words, the elements $\rho(R,R';\tau)$ of the density matrix [19, 42]. A path integral representation of this object can be obtained by formally bringing the number of imaginary time compositions in Eq. (5) to infinity [43], which yields

$$\rho(R,R';\tau) = \int_{X(0)=R}^{X(\tau)=R'} \mathcal{D}[X(t)]e^{-S[X(t)]}, \tag{23}$$

where the integration measure is

$$\mathcal{D}[X(t)] = \lim_{M\to\infty}\prod_{i=0}^{M}dX_i, \tag{24}$$

and

$$S[X(t)] = -\lim_{M\to\infty}\sum_{i=0}^{M}\log\left[\rho(X_i,X_{i+1};\delta\tau)\right] \tag{25}$$

is called the *total action* of path $X(t)$, with $X_i = X(i\times\delta\tau)$ and $\delta\tau = \tau/M$. Notice that all paths start at $R$ and end at $R'$ after an imaginary time $\tau$, which is effectively a boundary condition on this functional integration. These paths are indeed continuous, driven random walks in the $M\to\infty$, or conversely $\delta\tau\to 0$ limit.

In general, the challenge of computing these matrix elements stems from the fact that the Hamiltonian is almost always a sum of two or more non-commuting terms, such that it is impossible to calculate the matrix elements from each contribution individually. For the typical

case where $\hat{H}$ is a sum of a kinetic term plus a potential term, $\hat{H} = \hat{K} + \hat{V}$, this can be seen by noticing that

$$e^{-\tau\hat{H}} = e^{-\tau(\hat{K}+\hat{V})} = e^{-\tau\hat{K}}e^{-\tau\hat{V}}e^{\frac{\tau^2}{2}[\hat{K},\hat{V}]+\mathcal{O}(\tau^3)}, \tag{26}$$

with higher-order terms following the general recipe from the Baker-Campbell-Hausdorff formula [44–46]. However, this relation indicates that these contributions can be neglected in the limit $\tau \to 0$. In fact, the Trotter-Suzuki decomposition

$$e^{-\tau(\hat{K}+\hat{V})} = \lim_{M\to\infty}\left[e^{-\frac{\tau}{M}\hat{K}}e^{-\frac{\tau}{M}\hat{V}}\right]^M, \tag{27}$$

shows that, precisely in that limit, the contributions coming from the commutators vanish [47]. This is a rigorous mathematical result and makes complete sense physically. It represents the observed nature that, in the limit of very short $\tau$, which translates into either short time intervals or high temperatures, the system behaves classically [48].

Since the Trotter-Suzuki limit is intrinsically enclosed in each element of the total action in Eq. (25), we can rigorously separate $K$ and $V$ under the path integral exponentiation. The kinetic term can be analytically solved, having a well-known Gaussian distribution form, such that the elements become

$$\rho(X_i, X_{i+1}; \delta\tau) = \frac{1}{(4\pi\lambda\delta\tau)^{dN/2}}\exp\left[-\frac{(X_i - X_{i+1})^2}{4\lambda\delta\tau}\right]\exp[-\delta\tau V(X_i)], \tag{28}$$

where $d$ is the number of spatial dimensions. The Gaussian envelope from the kinetic contribution implicitly sets a relevant length scale over which the random walks have finite, non-vanishing probabilities, centered around the point where the interaction potential is being calculated and which is controlled in size by $\delta\tau$. These envelopes can be formally combined into the integration measure of Eq. (23), such that the density matrix elements are represented by a Wiener process through

$$\rho(R, R'; \tau) = \int_R^{R'} \mathcal{D}_W[X(t)]\exp\left[-\int_0^\tau V[X(t)]dt\right], \tag{29}$$

where $\mathcal{D}_W[X(t)]$ is the Wiener measure [49], which attributes statistical weights to each free-particle path starting at $R$ and ending at $R'$ after an imaginary time $\tau$.

The Wiener process expression of the density matrix describes a Brownian random walk (BRW) driven by the interaction term [50]. The functional integration can be written as an average over these random walks, in what is known as the Feynman-Kac formula [51]:

$$\rho(R, R'; \tau) = \rho_0\left\langle\exp\left[-\int_0^\tau V[X(t)]dt\right]\right\rangle_{BRW}, \tag{30}$$

where $\rho_0$ is the density matrix element for the non-interacting system,

$$\rho_0(R, R'; \tau) = \frac{1}{(4\pi\lambda\tau)^{dN/2}}\exp\left[-\frac{(R-R')^2}{4\lambda\tau}\right]. \tag{31}$$

Equation (30) defines a framework amenable to several approximations guided by physical intuition that will be discussed next. However, more common mathematical approaches can also be used, such as manipulating high-order commutators in the Baker-Campbell-Haussdorff expression [52–54], or general classical mechanics methods. One example is the use of van Vleck determinants to find trajectories with high stability against initial conditions [55, 56].

### 4.1 Short imaginary time approximations

#### 4.1.1 Primitive approximation

The most common, widely used expression for the density matrix considers that $\tau$ is small enough that the Brownian motion is effectively captured almost entirely by its initial and final states at $R$ and $R'$. This leads to the following expression, which is formally correct up to $\mathcal{O}(\tau^2)$,

$$
\begin{aligned}
\rho_{\text{pr}}(R,R';\tau) &= \rho_0 \, e^{-U(R,R',\tau)}, \\
U(R,R',\tau) &= \frac{\tau}{2}[V(R)+V(R')].
\end{aligned}
\tag{32}
$$

The symmetrized form of the potential action $U(R,R',\tau)$ in the primitive approximation is simply the interatomic potential. The primitive approximation is easy to compute during simulations, not requiring any operations related to the interaction potential and, therefore, being available to virtually every many-body system, apart from singular cases [57]. It performs exceptionally well for smooth, slowly varying potentials.

The pitfall here is that representing the entire range of the Brownian random walks exclusively by the endpoints is a pretty drastic restriction that can only be reasonable for *very* short imaginary time steps $\tau$. Consequently, it requires slicing the density operator into a sometimes prohibitively large number of chunks. This renders sampling the distribution of Eq. (16) a computationally demanding task in terms of time and resources. In practical terms, this means having to equilibrate, in a Markov process sense, many interacting polymers, each composed of a large number of beads. More accurate approximations are highly welcome for simulations to be efficient and, despite not being very common, can also be easily achieved.

#### 4.1.2 Semiclassical approximation

In many cases, it is possible to obtain a substantially better result by considering a WKB approach, which consists of finding the path amongst the Brownian random walks that has the largest statistical weight, denoted $X_{\text{WKB}}(t)$, and evaluating the interaction functional in Eq. (30) over that path. Aside from being physically intuitive, it can be rigorously shown that the path with the largest weight is exactly the classical path connecting $R$ to $R'$, justifying the name of the approximation. Moreover, since the BRWs are paths of the *non-interacting* system, $X_{\text{WKB}}(t)$ is simply a straight line

$$
\rho_{\text{WKB}}(R,R';\tau) = \rho_0 \exp\left[-\int_0^\tau V[X_{\text{WKB}}(t)]dt\right],
\tag{33}
$$

where $X_{\text{WKB}}(t) = R + (R'-R)t/\tau$.

The integration can be translated into a kinematics problem with a definite, analytical solution for some potentials. When that is not possible, direct numerical integration, with tabulation of the resulting terms, is a good alternative. This table can then be interpolated during simulations. In several situations, this is computationally faster than dealing with analytical forms, where intricate potentials must be evaluated repeatedly at every iteration.

By considering the interaction along the entire path connecting $R$ to $R'$, the semiclassical approximation can capture the physics of the system better than the primitive approach and should always be preferred whenever possible. However, since it is not derived directly from Baker-Campbell-Hausdorff terms, it is uncertain to what order of $\tau$ this approximation is correct, although we certainly know that it is at least $\mathcal{O}(\tau^2)$. Despite that, given a certain $\tau$, one can calculate the relative weight of $X_{\text{WKB}}$ to other possible random walks and thus have insights on how large $\tau$ should be to obtain the desired accuracy. These additional BRWs can be sampled with a simple Monte Carlo algorithm. The downside of this approximation is that, as

we shall see, some estimators require derivatives of the density matrix that may not be easily computed from this expression.

## 4.2 Pair product approximation

In a variety of quantum systems, the many-body physics is encapsulated into a two-body potential $v$, such that the interaction term of the Hamiltonian is

$$V(R) = \sum_{i<j} v(\mathbf{r}_i, \mathbf{r}_j), \tag{34}$$

where $(i,j)$ denotes a pair of particles. With that, we can write the many-body density matrix of Eq. (30) as

$$\rho(R,R';\tau) = \rho_0 \left\langle \prod_{i<j} \exp\left[ -\int_0^\tau v(\mathbf{r}_i(t),\mathbf{r}_j(t)) dt \right] \right\rangle_{BRW}. \tag{35}$$

The pair product approximation, $\rho_{\mathrm{PP}}$, consists of neglecting product correlations of orders higher than two in the Wiener process, such that

$$\left\langle \prod_{i<j} \exp\left[ -\int_0^\tau v(\mathbf{r}_i(t),\mathbf{r}_j(t)) dt \right] \right\rangle_{BRW} \approx \prod_{i<j} \left\langle \exp\left[ -\int_0^\tau v(\mathbf{r}_i(t),\mathbf{r}_j(t)) dt \right] \right\rangle_{BRW}, \tag{36}$$

and

$$\rho_{\mathrm{PP}}(R,R';\tau) = \prod_{i<j} \rho^{\mathrm{pair}}(\mathbf{r}_i,\mathbf{r}_j,\mathbf{r}_i',\mathbf{r}_j';\tau), \tag{37}$$

where

$$\rho^{\mathrm{pair}}(\mathbf{r}_i,\mathbf{r}_j,\mathbf{r}_i',\mathbf{r}_j';\tau) = \rho_0(\mathbf{r}_i,\mathbf{r}_j,\mathbf{r}_i',\mathbf{r}_j';\tau) \left\langle \exp\left[ -\int_0^\tau v(\mathbf{r}_i(t),\mathbf{r}_j(t)) dt \right] \right\rangle_{BRW}. \tag{38}$$

With that, the critical ingredient for the many-body density matrix is now the interaction term of *one pair* of particles, a much simpler object.

This approximation has several advantages. First, it is exact for a pair of particles by construction. Second, for homogeneous systems, the neglected correlation effects are largely attenuated since they tend to be canceled out by the presence of other particles in other directions that will have opposite correlations. Third, this approximation is well-defined for all sorts of interactions. Moreover, finally, for large imaginary times, this expression will approach the solution of a two-body Schrödinger equation (weighted by the corresponding energy eigenvalue), resulting in a many-body density matrix equivalent to a Jastrow-type wave function. These wave functions are well-known to accurately capture most ground state short-range correlations in systems that exhibit collective phenomena [30, 33, 34, 58, 59]. For systems composed of helium atoms, as stated by Leggett [60], the Jastrow function ansatz is the archetypal form of a variational ground state wave function. Corrections due to long-range correlations can be made based on quantizing the classical sound field and considering the zero-point motion of longitudinal phonons [61, 62]. Although this correction improves the accuracy of the static structure factor at small wave vectors, contributions to the energy are small because the interatomic potential is not long-range. Further improvements based on parametrical expansions are also possible [63]. In other systems, it may be very well the case where higher-order and long-range correlations are important, which must be addressed case-by-case.

#### 4.2.1 Central potentials

When the interaction is isotropic, depending only upon the magnitude $r$ of the relative coordinate of the pair of particles $(i,j)$, $r = |\mathbf{r}_i - \mathbf{r}_j|$, it is possible to employ a center of mass transformation such that the pair density matrix elements become

$$\rho^{\text{pair}}(\mathbf{r}_i, \mathbf{r}_j, \mathbf{r}'_i, \mathbf{r}'_j; \tau) = \rho^{\text{CM}}(\mathbf{r}_{\text{CM}}, \mathbf{r}'_{\text{CM}}; \tau)\rho^{\text{rel}}(\mathbf{r}, \mathbf{r}'; \tau), \tag{39}$$

where $\mathbf{r}_{\text{CM}} = (\mathbf{r}_i + \mathbf{r}_j)/2$ and $\mathbf{r}'_{\text{CM}} = (\mathbf{r}'_i + \mathbf{r}'_j)/2$ are the center of mass coordinates, and $\mathbf{r} = \mathbf{r}_i - \mathbf{r}_j$ and $\mathbf{r}' = \mathbf{r}'_i - \mathbf{r}'_j$ the relative coordinates at the endpoints. The center of mass density matrix $\rho^{\text{CM}}$ is effectively a free-particle one-body term on these coordinates, with the resulting mass $m_{\text{CM}} = m_i + m_j$. The relative density matrix $\rho_{\text{rel}}$ is the solution to the problem of a single particle of reduced mass $m_{\text{rel}} = m_i m_j/(m_i + m_j)$ under the action of an external potential $v(\mathbf{r})$.

In this scenario, one can immediately make use of one of the short imaginary time approximations to obtain an expression for $\rho^{\text{rel}}$, then reconstruct the pair density matrix $\rho^{\text{pair}}$, and finally obtain the many-body object with the pair product approximation. Alternatively, since this problem is much simpler than the many-body one, one can attempt to solve the one-body Schrödinger equation and construct an *exact* density matrix from

$$\rho^{\text{rel}}(\mathbf{r}, \mathbf{r}'; \tau) = \sum_i e^{-\tau E_i} \phi_i^*(\mathbf{r})\phi_i(\mathbf{r}'), \tag{40}$$

where $\phi_i$ are the eigenfunctions and $E_i$ the eigenvalues of the corresponding single-particle Hamiltonian.

### 4.3 Numerical convolutions

Accurate density matrices for imaginary times $\tau$ larger than the ones for which short-time approximations hold well are essential to reduce the total required number of beads in a simulation. They allow the PIGS method to project the system's ground state faster or with more particles with the same computational resources. Here, once again, the convolution property is quite convenient since we can use the approximations for short imaginary times and then self-compose them to obtain the expression for a larger imaginary time.

This is not practical for the full many-body matrix because it requires numerical integration of a highly multi-dimensional object. However, the central potential case discussed in the last section offers a great advantage: the relative coordinates term can be expanded in partial waves. In three dimensions, $\rho^{\text{rel}}$ becomes

$$\rho^{\text{rel}}(\mathbf{r}, \mathbf{r}', \tau) = \frac{1}{4\pi r r'} \sum_{\ell=0}^{\infty} (2\ell+1)\rho^{\ell}(r, r', \tau)P_{\ell}(\cos\theta), \tag{41}$$

where $\theta$ is the angle between $\mathbf{r}$ and $\mathbf{r}'$ and $P_{\ell}$ are Legendre polynomials. It can be shown that the partial waves $\rho^{\ell}$ are solutions of the one-body problem of a particle under an external potential $v(r)$ and a centrifugal barrier $\lambda_{\text{rel}}\ell(\ell+1)/r^2$, with $\lambda_{\text{rel}} = \hbar^2/2m_{\text{rel}}$. Each partial wave $\rho^{\ell}(r, r', \tau)$ in Eq. (41) is the solution to the following Bloch equation:

$$-\frac{\partial}{\partial t}\rho^{\ell}(r, r', t) = \left[-\lambda_{\text{rel}}\frac{d^2}{dr^2} + \lambda_{\text{rel}}\frac{\ell(\ell+1)}{r^2} + v(r)\right]\rho^{\ell}(r, r', t), \tag{42}$$

with boundary conditions $\rho^{\ell}(r, r', 0) = \delta(r, r')$ and $\rho^{\ell}(0, r', t) = 0$. For the free particle case, $v(r) = 0$, the solution is the free-particle contribution

$$\rho_0^{\ell}(r, r', \tau) = \frac{4\pi r r'}{(4\pi\lambda_{\text{rel}}\tau)^{3/2}} \exp\left[-\frac{(r^2 + r'^2)}{4\lambda_{\text{rel}}\tau}\right] i_{\ell}\left(\frac{r r'}{2\lambda_{\text{rel}}\tau}\right). \tag{43}$$

Finally, the partial waves $\rho^\ell$ for the three dimensional case can be written as

$$
\begin{aligned}
\rho^\ell(r,r',\tau) = {} & \frac{4\pi r r'}{[4\pi\lambda_{\text{rel}}\tau]^{3/2}} \exp\left[-\frac{(r^2+r'^2)}{4\lambda_{\text{rel}}\tau}\right] i_\ell\left(\frac{rr'}{2\lambda_{\text{rel}}\tau}\right) \\
& \times \left\langle \exp\left[-\int_0^\tau v(x(t))dt\right]\right\rangle_{\text{CBRW}},
\end{aligned}
\tag{44}
$$

where $i_\ell$ is the modified spherical Bessel function, and CBRW stands for centrifugal Brownian random walks. In the short imaginary time limit, the CBRW converges to the usual BRWs, which can be shown by employing an asymptotic expansion of $i_l$.

Since the partial waves are completely independent, each one of them satisfies its own convolution property,

$$
\rho^\ell(r,r',2\tau) = \int dr'' \rho^\ell(r,r'',\tau)\rho^\ell(r'',r',\tau),
\tag{45}
$$

which is now a *one-dimensional* integration. One can then consider imaginary times short enough such that an approximation (primitive or semiclassical) is reasonable, and then compute the partial waves with Eq. (44) and perform numerical integrations, each one effectively doubling the initial imaginary time and yielding the exact density matrix elements apart from numerical errors [64].

The starting temperature can be relatively high, such as $T \sim 10^3$ K for Helium systems. The integration is repeated until the desired temperature is reached. In the calculation of $\rho^{\text{rel}}$ using Eq. (41), the sum of partial wave components at the desired temperature is truncated when contributions to the sum are smaller than a chosen threshold $\epsilon$, typically $\epsilon \sim 10^{-8}$. As a bonus, the larger the imaginary time interval, the smaller the number of effectively contributing waves. For fixed $r$ and $r'$, contributions from partial waves with large angular momenta become increasingly irrelevant with increasing $\delta\tau$, which is a direct consequence of the spherical Bessel function weights.

Attempts to sample configurations that enclose larger relative distances become less rare at low temperatures since the free particle Gaussian distribution mainly dictates sampling. This distribution broadens as $\delta\tau$ increases. Such subtlety is at the heart of implementing FN-PIGS and other path-integral methods. Even if one can find an incredibly accurate density matrix at low temperatures, one must still employ a substantial number of beads in the simulation. Using few beads results in large displacements in the bisection algorithm, which tend to be rejected by the repulsive interaction part of the density matrix (particles tend to fall too close to others). The ideal average displacement, and therefore the associated value of $\delta\tau$ and the number of beads, is primarily controlled by the density of the system.

## 4.4 Construction of the many-body action

In practice, evaluating the action for a pair of particles, *viz.*

$$
u_{ij} = -\log\left[\frac{\rho^{\text{pair}}(\mathbf{r}_i,\mathbf{r}_j,\mathbf{r}_i',\mathbf{r}_j';\delta\tau)}{\rho_0(\mathbf{r}_i,\mathbf{r}_i',\delta\tau)\rho_0(\mathbf{r}_j,\mathbf{r}_j',\delta\tau)}\right],
\tag{46}
$$

is a costly task to be performed during simulation time, even when analytical expressions are available. A possible procedure is to store the $u_{ij}$ values in a table and form the many-body action via the pair product approximation, which yields

$$
U(R,R';\delta\tau) = \sum_{i<j} u_{ij},
\tag{47}
$$

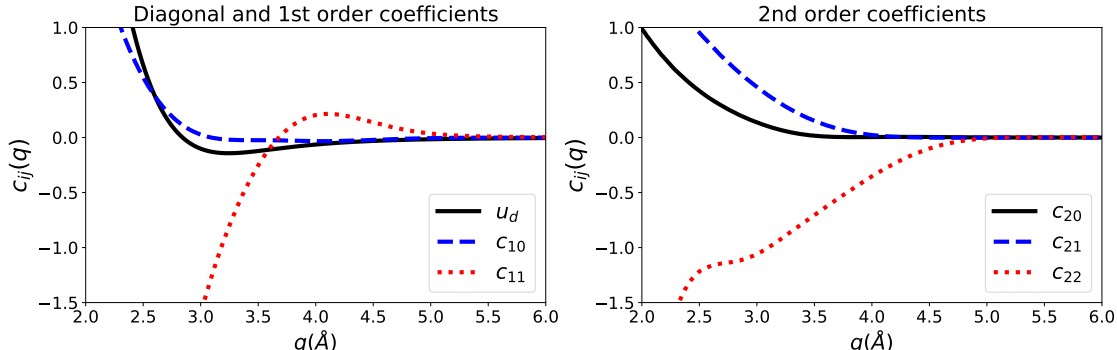

Figure 3: Diagonal pair action (black solid line, left panel), first-order coefficients (left panel), and second-order coefficients (right panel) of Eq. (48) for a pair $^3$He atoms interacting via the Aziz potential [65] at temperature $T = 50$ K obtained from six numerical convolutions of a short imaginary time WKB expression [see Eq. (33)].

and the full density matrix elements are $\rho(R,R';\delta\tau) = \rho_0(R,R';\tau)e^{-U(R,R',\delta\tau)}$. Even then, simply tabulating $u_{ij}$ as a function of distances $r$, $r'$ and the angle $\theta = \arccos(\mathbf{r}\cdot\mathbf{r}'/rr')$ for the required time intervals used in sampling $\mathcal{P}$ can generate very large four-dimensional arrays.

A much more efficient representation can be achieved by decomposing the pair action into diagonal and off-diagonal contributions [19], and writing $u_{ij}$ as

$$u_{ij}(r,r',\theta,\delta\tau) = u_{\text{ep}}(r,r',\delta\tau) + u_{\text{od}}(r,r',\theta,\delta\tau),\qquad(48)$$

where $u_{\text{ep}}(r,r',\delta\tau) = [u_{ij}(r,r,\delta\tau) + u_{ij}(r',r',\delta\tau)]/2$ is the so-called end-point term, and $u_{\text{od}}$ is the remaining contribution. The end-point term can be efficiently stored in tables since it is essentially a one-dimensional object.

The off-diagonal terms can be expanded by noticing that the density matrix elements are centered around the diagonal $\mathbf{r} = \mathbf{r}'$, a consequence of the Gaussian envelope that we discussed earlier in the path integral formalism. A suitable change of variables that encloses the proximity to the diagonal is given by $(r,r',\theta) \to (q,s,z)$, with $q = (r + r')/2$, $s = |\mathbf{r} - \mathbf{r}'|$ and $z = r - r'$. The variable $q$ is the only one that is not restricted by the Gaussian term [19]. With that, $u_{\text{od}}$ can be written as the following polynomial representation:

$$u_{\text{od}}(q,s,z,\delta\tau) = \sum_{i=1}^{\infty}\sum_{j=0}^{i} c_{ij}(q,\delta\tau)z^{2j}s^{2(i-j)}.\qquad(49)$$

This expression can be truncated, and the coefficients $c_{ij}$ are obtained by fitting a surface to $u_{\text{od}}$ for a set of values of $q$ that are relevant to the problem. The coefficients are then stored in one-dimensional tables that can be efficiently accessed during simulations, allowing for quick construction of the many-body action. For Helium systems, $i = 2$ is an adequate truncation point. Figure 3 shows the resulting coefficients for a pair of $^3$He atoms interacting via the Aziz potential [65] at temperature $T = 50$ K.

## 5 Sampling algorithm

In the PIGS method, sampling the relevant probability distributions can be done with stochastic techniques. However, for polymers with more than 16 beads [19], detailed balance is challenging to achieve when one relays on the canonical Metropolis algorithm and its variants

based on configuration-by-configuration sampling [66, 67]. The multi-level Metropolis algorithm speeds up the sampling process and is, therefore, preferred [19]. This algorithm divides the sampling into stages (levels). In the initial stages, a crude but fast approximation for the probability density is used to decide if the proposed moves are accepted. In the final stage, the most accurate expression of the density matrix is employed, and its associated probability density is sampled for all beads.

The idea is to use more simple forms of the density matrix during the early stages to filter trial moves with a reasonable probability of being accepted in the final stage. It is undesirable to spend simulation time evaluating sophisticated expressions of the density matrix before the last stage. For example, one can use the primitive approximation to discard situations where hard-core particles overlap. The more accurate pair density matrix expressions can be used at the final stage.

The multi-level Metropolis algorithm implementation we employ is known as the bisection algorithm. In a bisection of level $L$, a segment $\mathcal{R} = \{R_i \mid i, \ldots, i + \mathcal{N}\}$ with $2^L + 1$ beads of the open necklace is randomly selected. If $(i + \mathcal{N}) > M$, the segment is discarded, and another one is selected. At the first stage, $\ell = 1$, a trial configuration $R'_m$ is proposed for the central bead $R_m$ of the segment delimited by $(R_i, R_{i+\mathcal{N}})$, $\mathcal{N} = 2^L$. If the proposed move in this stage or any subsequent one is rejected, a different segment $\mathcal{R}$ is selected, and the iteration is restarted. When the trial configuration $R'_m$ is accepted, the algorithm proceeds to the next stage. At stage $\ell = 2$, the chosen segment is split in two with extremities $(R_i, R'_m)$ and $(R'_m, R_{i+\mathcal{N}})$. Each one includes the updated configuration $R'_m$ of the previous stage. A new configuration for the central bead of each one of these two new segments is then proposed. Although the proposed moves are independently generated, the criterion for their acceptance, as we will see in Eq. (54), considers all of them simultaneously. If the proposed moves of the centers in this stage are accepted, the algorithm proceeds to the next stage. The same recipe is repeated recursively in the following stages, considering bisections of segments constructed in the prior stages. $2^{\ell-1}$ segments are to be considered at each stage. In Fig. 4, we show a graphical representation of the bisection algorithm with $L = 2$.

A segment with extremities $(R_{e1}, R_{e2})$ has its central bead $R_m$ separated by an imaginary time $\delta\tau_\ell = 2^{L-\ell}\delta\tau$ from the endpoints. The probability density associated with this segment is

$$P_\ell(R_{e1}, R_{e2}) = \rho_0(R_{e1}, R_m, \delta\tau_\ell)\rho_0(R_m, R_{e2}, \delta\tau_\ell)e^{-\mathcal{U}_\ell}, \tag{50}$$

with the action $\mathcal{U}_\ell$ given by a single sum over all the segments $\mathcal{S} = R_{e1} - R_m - R_{e2}$ considered at the present stage,

$$\mathcal{U}_\ell = \sum_{\mathcal{S}} \left[ U(R_{e1}, R_m, \delta\tau_\ell) + U(R_m, R_{e2}, \delta\tau_\ell) \right]. \tag{51}$$

The product of the free-particle density matrices in Eq. (50) is proportional to

$$\exp\{-[R_m - (R_{e1} + R_{e2})/2]^2/(4\lambda\delta\tau_\ell)\}, \tag{52}$$

and therefore can be exactly sampled with the trial displacement

$$R_m^{\text{trial}} = \frac{R_{e1} + R_{e2}}{2} + \eta\sqrt{2\lambda\delta\tau_\ell}, \tag{53}$$

where $\eta$ is a vector of random numbers generated from a normal distribution of unitary variance and zero mean. This trial configuration is accepted or rejected by the Metropolis algorithm according to the acceptance probability

$$A_\ell = \min\left\{1, \frac{e^{-\mathcal{U}_\ell^{\text{trial}} + \mathcal{U}_\ell}}{e^{-\mathcal{U}_{\ell-1}^{\text{trial}} + \mathcal{U}_{\ell-1}}}\right\}, \tag{54}$$

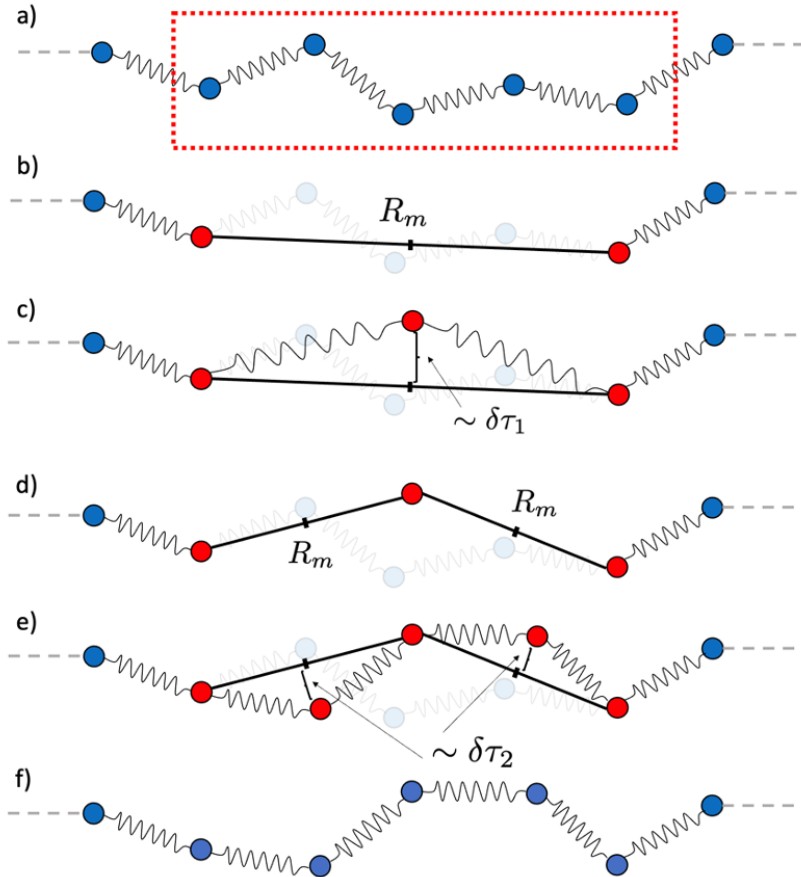

Figure 4: Visual representation of the bisection algorithm of level $L = 2$. **a)** A random slice of the open necklace with $2L + 1 = 5$ beads is selected. **b)** The endpoints of the selected segment are kept fixed, and we calculate the midpoint between them, $R_m$. **c)** A trial displacement is proposed via Eq. (53), effectively sampling a Gaussian distribution controlled by $\delta\tau_1 = 2^{2-1}\delta\tau = 2\delta\tau$. **d)** The midpoints between the new position of the bead in the previous step and the endpoints are calculated. **e)** Trial displacements are then proposed via Eq. (53) with $\delta\tau_2 = 2^{2-2}\delta\tau = \delta\tau$. **f)** If the movements in all levels are accepted, we update the positions of the beads, as discussed in the text.

where $\mathcal{U}_\ell^{\text{trial}}$ is obtained by taking $R_m = R_m^{\text{trial}}$ in $\mathcal{U}_\ell$, with $\mathcal{U}_0 = \mathcal{U}_0^{\text{trial}} = 0$ by definition [67].

When calculating $\mathcal{U}_\ell^{\text{trial}}$ at any stage, we consider the updated configurations of all previous stages. The original configurations of the segment are used to compute $\mathcal{U}_\ell$. For $\ell < L$, $\mathcal{U}$ is calculated from the primitive approximation.

The algorithm proceeds to the next stage if the trial move is accepted. Otherwise, the entire segment is left unchanged, a new one is selected, and the iteration is restarted. In the last stage, $\ell = L$, each segment is composed of three beads. Except for the endpoints $R_i$ and $R_{i+\mathcal{N}}$ and those at the center of all segments with three beads, all others have configurations from previously accepted moves. The imaginary time interval to be employed in this stage is the one used for calculating the pair density matrix, and $\mathcal{U}$ is calculated using the more refined pair product expression. If accepted, bead configurations are updated as $\{R_i = R_i' \mid i + 1, \ldots, i + \mathcal{N} - 1\}$; otherwise, $\mathcal{R}$ remains unchanged. Since configurations of beads $R_i$ and $R_{i+\mathcal{N}}$ are never changed, the procedure of choosing a segment $\mathcal{R}$ needs to be repeated several times for the same polymer.

In the final stage, the denominator in Eq. (54) cancels the approximated probability densities used in the previous stages. The final acceptance probability $A = \prod_\ell A_\ell$ is how the most accurate expression of the many-body density matrix for accepting moves of all bead configurations is obtained. For this reason, the movements at all levels must be either all accepted or all rejected. In this way, the detailed balance required for an unbiased estimate of properties is satisfied.

Generally, acceptance is controlled by the imaginary time interval $\delta\tau$ and the total number of stages $L$. For bosonic systems, the usual acceptance is $\sim 20\%$. However, a finite probability of having a cross-recross error exists for fermionic systems, where the fixed-node approximation is employed. This error occurs when a trial movement crosses two nodal surfaces, reaching pockets with the same wave function sign. To avoid this error, we monitor the number of attempted moves that result in a sign change of the trial wave function and are discarded due to the fixed node approximation. This number, for acceptance ratios above 90%, represents less than 0.4% of the total attempted moves in a simulation, and it is even smaller, 0.02%, for acceptance ratios over 99%. In this scenario, having a cross-recross error due to a large displacement is improbable. This unlikelihood is reflected in the fact that ground state energies obtained with acceptance ratios above 90% statistically agree among themselves. However, higher acceptance rations imply much longer simulation times. Thus, 90% is a recommended value.

A Monte Carlo sweep is finally completed with attempts to move the configurations that carry wave functions $\Psi_T$ at the extremities of the open necklace. Trial configurations are generated according to the Metropolis algorithm [66,67]. The canonical algorithm is associated with a step size that regulates the average displacement of the particles and the acceptance ratios. This step size is also adjusted to give an overall acceptance ratio of $\gtrsim 90\%$ when the fixed node approximation is used.

Over the course of decades, extensive variational studies provide a collection of suitable trial functions related to various physical systems. Nonetheless, the art of constructing these functions is still an active field, including machine learning techniques [68–74] and quantum computing [75]. It is essential to mention that accurate trial functions expedite simulations in the sense that shorter projections, resulting in fewer beads, will be necessary to reach convergence.

Most of our simulations were performed with an $L = 3$ bisection algorithm. In this case, the number of contiguous beads selected to be treated independently is 7, the two extremities of the segment are kept fixed. The total number of beads in each polymer usually considered for the density matrix is typically a few times larger. Therefore, in our simulations, a Monte Carlo sweep consists of several applications of the bisection algorithm until, on average, all beads are given a chance to have updated configurations.

## 6 Estimators

The PIGS method paves the way to evaluate the expected values of several properties of physical systems at $T = 0$. Unbiased estimates for local properties, such as the total, kinetic and potential energies, and the radial distribution function, can be obtained by constructing estimators from the corresponding operators. In particular, operators in coordinate representation have a direct estimator expression that follows from Eqs. (14) and (15). Once the sampling algorithm has achieved an equilibrium state, the configurations of the polymers are used to calculate a Monte Carlo ground state estimate $O(\tau)$ associated with operator $\hat{O}$ following the expression

$$O(\tau) = \langle O(R_0, R_1) \rangle_{\mathcal{P}} , \qquad (55)$$

where $\langle\ldots\rangle_{\mathcal{P}}$ denotes the average over uncorrelated configurations sampled from the probability distribution of Eq. (16) and $O(R_0, R_1)$ is given by Eq. (15).

## 6.1 Densities and radial distribution function

To obtain an operator expression for the radial distribution function, we initially consider the one-body number density operator, which is defined in terms of field operators as

$$\hat{d}^{(1)}(\mathbf{r}) = \hat{\Psi}^{\dagger}(\mathbf{r})\hat{\Psi}(\mathbf{r}). \tag{56}$$

Any state vector in the coordinate representation of a system of $N$ particles can be written as

$$|R\rangle = \bigotimes_{j=1}^{N}|\mathbf{r}_j\rangle, \tag{57}$$

such that the action of field operators in these states is given by

$$\hat{\Psi}^{\dagger}(\mathbf{r})|R\rangle = \hat{\Psi}^{\dagger}(\mathbf{r})\bigotimes_{j=1}^{N}|\mathbf{r}_j\rangle = |R+\mathbf{r}\rangle, \tag{58}$$

where $|R+\mathbf{r}\rangle$ correspond to the configuration $R$ with the addition of one particle at $\mathbf{r}$, and

$$\hat{\Psi}(\mathbf{r})|R\rangle = \hat{\Psi}\bigotimes_{j=1}^{N}|\mathbf{r}_j\rangle = \sum_{j=1}^{N}\delta(\mathbf{r}-\mathbf{r}_j)|R-\mathbf{r}\rangle, \tag{59}$$

where $|R-\mathbf{r}\rangle$ correspond to the configuration $R$ with the annihilation of one particle at $\mathbf{r}$. Naturally, this only makes sense if there *is* one particle at $\mathbf{r}$ in the configuration $R$, which is where the delta functions stem from.

We can see then that $\hat{d}^{(1)}(\mathbf{r})$ is a local operator, since

$$\hat{\Psi}^{\dagger}(\mathbf{r})\hat{\Psi}(\mathbf{r})|R\rangle = \hat{\Psi}^{\dagger}(\mathbf{r})\sum_{j=1}^{N}\delta(\mathbf{r}-\mathbf{r}_j)|R-\mathbf{r}\rangle = \sum_{j=1}^{N}\delta(\mathbf{r}-\mathbf{r}_j)|R\rangle, \tag{60}$$

therefore its estimator is given by

$$d^{(1)}(\mathbf{r};\tau) = \left\langle \frac{\langle R_0|p(R_0,R_1)\hat{\Psi}^{\dagger}(\mathbf{r})\hat{\Psi}(\mathbf{r})|R_1\rangle}{p(R_0,R_1)} \right\rangle_{\mathcal{P}} = \left\langle \sum_{j=1}^{N}\delta(\mathbf{r}-\mathbf{r}_j^{(0)}) \right\rangle_{\mathcal{P}_{\mathrm{loc}}}, \tag{61}$$

where $\mathbf{r}_j^{(0)}$ is the position of particle $j$ in $R_0$, and the average is over the probability distribution $\mathcal{P}_{\mathrm{loc}}$ from Eq. (18). For a uniform liquid, this final expression is simplified to $d^{(1)}(\mathbf{r};\tau) = N/V$, where $V$ is the volume of the system.

We can use the same approach to calculate the two-body number density operator, which is written in terms of field operators as

$$\hat{d}^{(2)}(\mathbf{r}_1,\mathbf{r}_2) = \hat{\Psi}^{\dagger}(\mathbf{r}_1)\hat{\Psi}^{\dagger}(\mathbf{r}_2)\hat{\Psi}(\mathbf{r}_2)\hat{\Psi}(\mathbf{r}_1), \tag{62}$$

with the resulting estimator given by

$$d^{(2)}(\mathbf{r}_1,\mathbf{r}_2;\tau) = \left\langle \sum_{i=1}^{N}\sum_{\substack{j=1\\j\neq i}}^{N}\delta(\mathbf{r}_1-\mathbf{r}_i^{(0)})\delta(\mathbf{r}_2-\mathbf{r}_j^{(0)}) \right\rangle_{\mathcal{P}_{\mathrm{loc}}}. \tag{63}$$

Finally, the radial distribution function operator, defined as

$$\hat{g}(\mathbf{r}_1, \mathbf{r}_2) = \frac{\hat{d}^{(2)}(\mathbf{r}_1, \mathbf{r}_2)}{d^{(1)}(\mathbf{r}_1) d^{(1)}(\mathbf{r}_2)}, \tag{64}$$

has, for a uniform liquid with spherical symmetry, the following estimator:

$$g(r) = \frac{V}{N^2} \left\langle \sum_{i=1}^{N} \sum_{j=1, j \neq i}^{N} \delta(r - |\mathbf{r}_i^{(0)} - \mathbf{r}_j^{(0)}|) \right\rangle_{\mathcal{P}_{\text{loc}}}, \tag{65}$$

where $r = |\mathbf{r}_1 - \mathbf{r}_2|$.

For systems with two spin components, $g(r)$ can be further separated in different manners. A simple one is in up-and-down spins

$$g_{\uparrow\uparrow, \uparrow\downarrow} = \frac{1}{N\rho} \left\langle \sum_{i,j}^{i \neq j} \frac{1 \pm \sigma_z(i)\sigma_z(j)}{2} \delta(r_{ij} - r) \right\rangle_{\mathcal{P}_{\text{loc}}}, \tag{66}$$

where the plus and minus signs correspond to $g_{\uparrow\uparrow}$ and $g_{\uparrow\downarrow}$, respectively.

## 6.2 Total energy

Total energies can be calculated in several ways by identifying $\hat{O}$ with the Hamiltonian of the system. If we follow the same recipe as the one used for the radial distribution function, we can show that $\hat{H}$ is local in coordinate representation. The estimator expression, also known as the *direct estimator*, is given by

$$E(\tau) = \left\langle V(R_0) - \frac{\lambda}{\rho(R_0, R_1; \delta\tau)} \frac{\partial^2}{\partial R_0^2} \rho(R_0, R_1; \delta\tau) \right\rangle_{\mathcal{P}_{\text{loc}}}, \tag{67}$$

where the derivative represents a Laplacian over all particle coordinates of $R_0$. This estimator is not convenient since it involves second derivatives of density matrix elements.

### 6.2.1 Thermodynamic estimator

It can be more convenient to consider a different approach to estimate the total energy. Since $\hat{\rho}(\delta\tau) = \exp(-\delta\tau\hat{H})$, we can write

$$\hat{H}\hat{\rho}(\delta\tau) = -\frac{\partial}{\partial(\delta\tau)} \hat{\rho}(\delta\tau). \tag{68}$$

Therefore, the expected value of the total energy can be estimated via the so-called thermodynamic estimator

$$E_{\text{th}}(\tau) = \left\langle -\frac{\partial}{\partial(\delta\tau)} \log \rho(R_0, R_1; \delta\tau) \right\rangle_{\mathcal{P}_{\text{loc}}}. \tag{69}$$

This estimator has the advantage of requiring only a first-order, non-positional derivative of the density matrix.

### 6.2.2 Mixed estimator

Given the fact that $[\hat{H}, \hat{\rho}] = 0$, $\hat{H}$ does not need to be estimated in the central bead configuration $R_0$. It is possible to estimate the system's total energy by applying the energy estimator to each one of the internal beads of the open polymer and, as a consequence, attenuate the statistical error associated with the estimation process.

It is possible to move $\hat{H}$ to one of the endpoints of the open necklace, which carries the trial wave function $\Psi_T$, such that

$$E(\tau) \propto \langle \Psi_T | e^{-\tau\hat{H}} \hat{H} e^{-\tau\hat{H}} | \Psi_T \rangle = \langle \Psi_T | e^{-2\tau\hat{H}} \hat{H} | \Psi_T \rangle . \tag{70}$$

This leads to the ubiquitous mixed estimator of the Diffusion Monte Carlo (DMC) method [76]. The mixed estimator allows for the total energy of the system for a given configuration $R_M$ to be estimated without bias through the so-called local energy

$$E_{\text{mix}}(\tau) = \left\langle \frac{H\Psi_T(R_M)}{\Psi_T(R_M)} \right\rangle_{\mathcal{P}_{\text{loc}}} , \tag{71}$$

where $H = -\lambda \frac{\partial^2}{\partial R_M^2} + V(R_M)$.

Despite not being essential as in the DMC method, the total energy mixed estimator is beneficial in the PIGS method. The total energy can be estimated with half the number of beads necessary to estimate quantities that do not commute with $\hat{H}$ because filtering the ground state is only required from one of the extremities of the open necklace, and the Hermiticity of $\hat{H}$ can be employed. Thus, this estimator offers a practical way to determine how many beads are needed in a given simulation to obtain converged configurations at the central bead, where most quantities are calculated.

It is important to mention some peculiarities of the mixed estimator. Suppose that the trial wave function $\Psi_T$ is not orthogonal to any of the true eigenstates of the system. In that case, the estimator will lead to vanishing variances of total energy estimates when $\Psi_T$ approaches the true eigenstate. Nevertheless, if $\Psi_T$ is not precisely the true eigenstate, a quantity $O$ that does not commute with the Hamiltonian requires an extrapolation, $O = 2O_{\text{mix}} - O_{\text{var}}$, where $O_{\text{var}}$ is the variational estimate obtained with $\Psi_T$ [25]. When a simulation displays a zero-variance situation, the true wave function is known. In this case, the obvious approach is to perform a simple Monte Carlo simulation to sample the probability distribution directly if integrations cannot be performed analytically.

### 6.3 Potential energy

The potential energy can be immediately estimated using a direct estimator. A simple calculation gives us the expression

$$V(\tau) = \langle V(R_0) \rangle_{\mathcal{P}_{\text{loc}}} , \tag{72}$$

which is straightforward to calculate during simulations.

### 6.4 Kinetic energy

If the potential energy operator $\hat{V}$ is independent of the single particle mass $m$, one can estimate the kinetic energy by considering the expression

$$\hat{K}\hat{\rho}(\delta\tau) = \frac{m}{\delta\tau} \frac{\partial}{\partial m} \hat{\rho}(\delta\tau) \tag{73}$$

to construct the thermodynamic estimator of $\hat{K}$, which is then given by

$$K_{\text{th}}(\tau) = \left\langle \frac{m}{\delta\tau} \frac{\partial}{\partial m} \log \rho(R_0, R_1; \delta\tau) \right\rangle_{\mathcal{P}_{\text{loc}}} . \tag{74}$$

A local estimator of the kinetic energy can also be constructed by directly using the quantum mechanical operator associated with this quantity. Nevertheless, it involves the Laplacian of

the density matrix $\rho(R_0, R_1; \delta\tau)$, which can be numerically unstable and have a high computational cost. In practice, to estimate the kinetic energy, the most accessible approach is to consider the straightforward estimation of the potential energy subtracted from the total energy estimate.

## 7 Results

As a proof-of-concept, the harmonic oscillator helps clarify several aspects involved in implementing the PIGS method for states described by symmetrical and anti-symmetrical wave functions. We then discuss $^4$He atoms as a paradigm of strongly correlated many-body systems. It offers the possibility of showing characteristics of the PIGS method in its simplest form. Conversely, results for bulk $^3$He show how our extension for Fermi particles can be applied in this prototype many-body system, establishing the basis for the treatment of any fermionic system. Finally, we show results for several $^3$He concentrations in the $^3$He-$^4$He mixture.

### 7.1 The harmonic oscillator as a proof-of-concept

As an initial application of the PIGS method, we employ a particle in a one-dimensional harmonic potential well in units of $m = \omega = \hbar = 1$. The ground state of the system under these conditions has energy $E_0 = 1/2$ and is associated with an even wave function $\psi_0(x)$, *i.e.*, $\psi_0(-x) = \psi_0(x)$, $\forall x \in \Re$. To verify the properties of the PIGS method, we choose a symmetrical trial state $\Psi_T(x) = \exp(-bx^2)$, where we assume $b \neq 1/2$. Although the functional form of $\Psi_T$ is the same as of the true ground state, $b \neq 1/2$ guarantees only a partial superposition of $\Psi_T(x)$ with $\psi_0(x)$ as long as $\Psi_T(x)$ is not orthogonal to the ground state. In our calculations, the parameter $b$ was set to 1.1, which yields a variational energy of $0.665 \pm 0.003$. This value can be obtained in a standard variational Monte Carlo calculation or equivalently by imposing $\tau = 0$ in the PIGS method. Of course, the exact value could also be calculated analytically.

We tested two situations for the initial coordinates of the polymers: all beads starting at the origin and each bead starting at a different, random position. No difference in overall performance was observed between these two initial conditions. We attribute this to the simplicity of the system. The imaginary time evolution to the ground state was performed using $\delta\tau = 0.1$ and the primitive approximation of Eq. (32) for the density matrix. These simulations were performed with the bisection algorithm using an $L = 3$ level. In a conventional PIGS run, we assigned 5% of the computational time to equilibration or thermalization to ensure detailed balance is achieved, with the remaining time used to estimate quantities of interest. Block averages of all quantities are aggregated to avoid correlations in the calculations of variances, and estimated errors are computed after the system has reached equilibrium.

The energy of the system was calculated using both the thermodynamic and the mixed estimator, Eq. (69) and Eq. (71), respectively. Both estimators give unbiased estimates, subjected only to statistical uncertainties. However, for small $\delta\tau$ values and considering a single bead, the energy standard errors associated with the thermodynamic estimator are larger than those corresponding to the mixed estimator. The larger error bars are a consequence of derivatives of the density matrix with respect to $\delta\tau$, which includes a narrow Gaussian for small values of $\delta\tau$. A ground state energy estimate of $0.50 \pm 0.01$ was obtained with the thermodynamic estimator, in excellent agreement with the analytical result $E_0 = 1/2$. For the mixed estimator, in Fig. 5, we show the evolution of the results towards the ground state energy as the imaginary time increases. The convergence to the ground state is observed with $\tau = 2$, *i.e.* $M = 20$ convolutions of the density matrix.

The first excited state of the harmonic oscillator is associated with an odd wave function $\psi_1(x)$, *i.e.*, $\psi_1(-x) = -\psi_1(x)$, $\forall x \in \Re$, which divides the $x$-axis into two regions of opposite

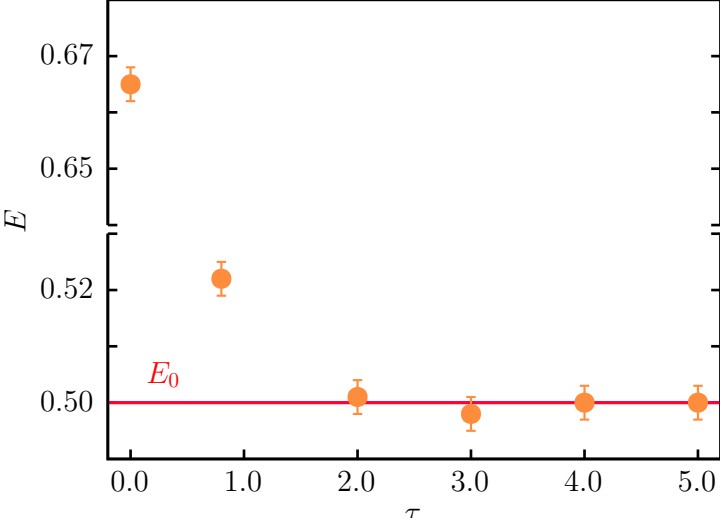

Figure 5: Evolution to the ground state energy of the harmonic oscillator obtained using the mixed estimator. Dots represent the average energy as a function of the imaginary time $\tau = 2M\delta\tau$. The solid line represents the analytical value of the ground state energy $E_0 = 1/2$. $E(\tau = 0)$ is the variational value.

signs, similar to the nodal regions of fermionic systems. For this reason, this state was chosen as a proof of concept to study the behavior of the PIGS method when it is mandatory to handle a state described by an anti-symmetric wave function.

The choice of the trial state $\Psi_T(x) = x\exp(-bx^2)$, orthogonal to the ground state $\psi_0(x)$, allows the imaginary time evolution to converge to the first excited state. The variational energy obtained by setting the parameter $b = 1.1$ in this trial function is $E(\tau = 0) = 1.992 \pm 0.006$. $E_1 = 3/2$ is the exact value.

In this case, the fixed-node approximation must be considered if we expect a reliable description of the properties of the system, especially its energy. This approximation was performed by restricting the sampling algorithm to positive values of $x$, and employing the primitive density matrix approximation of Eq. (32) along with the image action correction of Eq. (21). Results are displayed in Fig. 6. We performed simulations without the image action correction to showcase its importance. The simulation details were similar to those of the ground state energy estimates. However, without the image action correction, a much shorter imaginary time step $\delta\tau = 0.01$ was needed to obtain accurate results.

The imaginary time evolution towards the first excited state using the mixed estimator can be seen in Fig. 6. Results are notably better when the image action correction, represented by the circles, is used. Convergence is obtained for $\tau = 2$. Even though the results obtained with and without the image action correction are indistinguishable within a 99% confidence interval, the introduction of the image action correction systematically improves the results and reduces the computational cost of each simulation. For $\tau = 5.0$, the estimated energy with the image action correction is $1.51\pm0.02$, which is in excellent agreement with the first-excited state energy value. For calculations without the image action correction, the estimated energy is $1.56\pm0.02$. The oscillatory behavior of the results is related to the vanishing of probabilities in the vicinity of the nodal surfaces not being well captured without the correction.

Since the density matrix of the harmonic oscillator is known exactly [77], features of the PIGS method related to properties of the thermodynamic and mixed estimators for the total

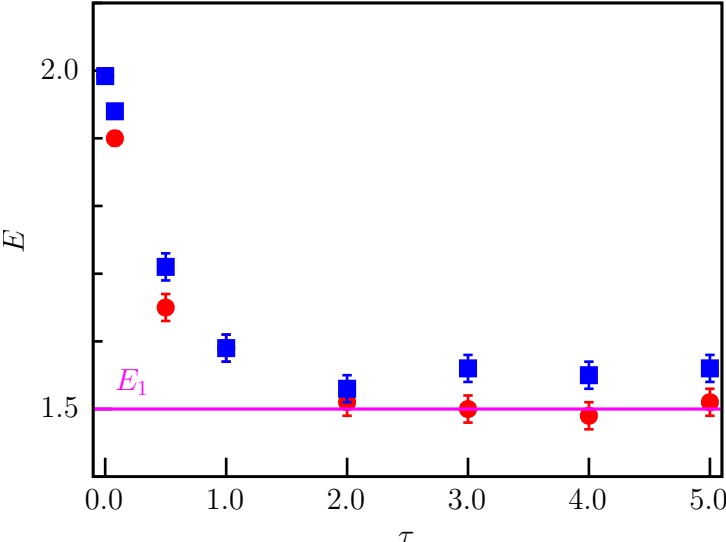

Figure 6: Evolution of estimates of the first-excited state energy for the harmonic oscillator as a function of the imaginary time $\tau$. The circles stand for results when the image action is employed. The squares represent these quantities without using the image action correction in the simulations. Error bars, when not visible, are smaller than the size of the symbol. The mixed estimator was used in both cases. The line shows the analytical value of the first excited-state energy $E_1 = 3/2$.

energy can be further investigated. The exact action,

$$U(x, x', \tau) = \frac{(x-x')^2}{4\lambda\tau} - \frac{1}{2}\ln\left[2\pi\sinh(\tau)\right] - \frac{1}{2\sinh(\tau)}\left[(x+x')^2\cosh(\tau) - 2xx'\right], \quad (75)$$

allows large $\tau$ values to be used without requiring convolutions.

The exact action is straightforward for the ground state, where we do not apply the fixed node approximation. A single bead is enough if the mixed estimator is applied to one of the wave functions at the extremities. The thermodynamic estimator was used only at the central bead of a necklace formed by three beads. This choice allowed converged configurations to the ground state at both sides of the central bead. It offered a hint of the difficulties encountered in estimating quantities that do not commute with the Hamiltonian of the system. We projected the ground state from the same even trial wave function used in the calculations with the primitive approximation. Estimates of the total energy using only configurations from the central bead as a function of $\tau$ are presented in Fig. 7. As already mentioned, for small $\tau$ values, the error bars associated with the thermodynamic estimator of the energy applied only to the central bead are considerably larger than the errors associated with the mixed estimator. Nevertheless, in this case, using the exact density matrix of the harmonic oscillator and with a fixed number of beads, the error bars decrease as $\tau$ increases since the derivatives of the density matrix with respect to $\tau$ include a broadening Gaussian. After convergence ($\tau = 5.0$), the average energies obtained were $E_{\text{th}} = 0.4999 \pm 0.0002$ for the thermodynamic estimator and $E_{\text{mix}} = 0.499 \pm 0.003$ for the mixed estimator.

When the fixed-node approximation is used to study the first excited state, which requires an infinite positive potential for $x \leq 0$, the action of Eq. (75) is no longer the exact action of the problem. As a result, a small value of $\delta\tau$ and convolutions of the density matrix is necessary for convergence. We projected the first excited state from the same odd trial wave function used in the calculations with the primitive approximation and performed simulations

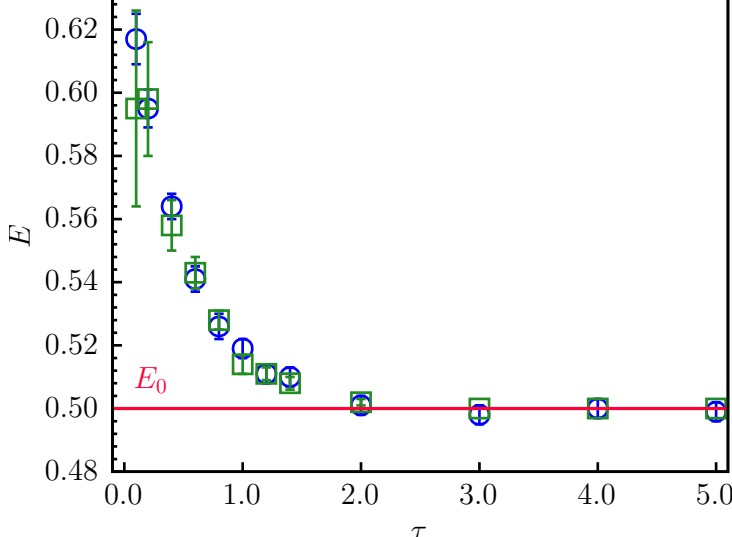

Figure 7: Evolution of the total energy to the ground state of the harmonic oscillator as a function of the imaginary time $\tau$ using the exact action. A single bead is enough for estimates using the mixed estimator, whose values are represented by squares. Results obtained through the thermodynamic estimator at the central bead of a three beads necklace are displayed by circles. The line shows the analytical value of the ground state energy $E_0 = 1/2$.

with and without the image action correction. In the former case, after convergence, the average energy obtained with the mixed estimator was $1.56 \pm 0.02$. When applied to the single central bead, a larger error bar is associated with the thermodynamic estimator, resulting in $1.38 \pm 0.17$. These results were improved when the image action correction was considered, with the thermodynamic estimator giving the energy $1.496 \pm 0.007$ and the mixed estimator giving $1.502 \pm 0.006$. In this simple model, the location of the node of the trial function is precisely known. Therefore, the true distance to the nodal surface, without approximation, was used to calculate the image action. For this reason, the convergence is to the true ground state energy.

## 7.2 Liquid helium

A complete description of mixtures of helium atoms is given by the Hamiltonian

$$H = -\sum_\alpha \sum_i^{N_\alpha} \frac{\hbar}{2m_\alpha} \nabla_i^2 + \sum_{i<j} v(r_{ij}), \tag{76}$$

where $m_\alpha$ is the isotope mass and $N_\alpha$ is the number of atoms of each species. In most computer simulations, the two-body inter-atomic potential is proposed by Aziz and collaborators [78]. In our simulations, we use a Hartree-Fock damped form that mimics the entire configuration interaction in the intermediate-range, the HFD-B3-FCI1 potential [65], which gives excellent results in the short-, medium-, and long-range regions. The $i < j$ sum in the interacting potential is over the total number of atoms in the system. Simulations of pure isotopes are performed by omitting the sum in the $\alpha$ index.

In studying isotope mixtures of helium atoms, it is interesting to have additional results for systems made from pure $^3$He and $^4$He atoms. Moreover, the PIGS method applied to a

many-body bosonic system, where the difficulties associated with fermionic systems are not present, helps clarify practical aspects of the method.

### 7.2.1 Pure $^4$He

The $^4$He trial function adopted in the extremities of the necklace was chosen to be of the Jastrow form,

$$\Psi_J(R) = \prod_{i<j} f(r_{ij}).\tag{77}$$

The factor $f(r_{ij}) = \exp[-u(r_{ij})/2]$ explicitly correlates pairs of particles through a pseudo-potential of the McMillan form $u(r_{ij}) = (b/r_{ij})^5$, where $b$ is a parameter [79]. Certainly, more sophisticated wave functions will favor a reduction in the number of beads considered for converged results and, consequently, make for a faster simulation.

Our simulations were performed with 108 $^4$He atoms at density $\rho = 0.02186\,\text{Å}^{-3}$. A convolution of the density matrix with 20 beads would produce converged configurations of the system ground state when the many-body action developed in Section (4.4) is considered. In other words, results obtained from a necklace of 41 beads guarantee that the central bead will have converged ground state configurations to estimate any property. Similar results can be obtained if the primitive approximation is used instead. However, more beads will be necessary to reach convergence. The equilibration stage was performed using 5% of the computational time of the run, with the remaining time employed in production stages. Again, block averages of all the quantities of interest are formed to avoid statistical correlations.

The total ground state energy of the system obtained through the thermodynamic estimator of Eq. (69) applied to all beads of the open polymer gives $E_{\text{th}} = -7.36 \pm 0.03$ K. It is also possible to estimate the total energy by applying the mixed estimator of Eq. (71) to the trial functions, with the result $E_{\text{mix}} = -7.31 \pm 0.01$ K in agreement with the thermodynamic estimator result. An experimental value [80] obtained at 1.3 K gives a total energy of $-7.17$ K. These values are summarized in Table 1. These results agree with other implementations of the PIGS method [30, 33]. In the literature, different two-body interatomic potentials that may or may not consider three-body effects in an effective manner [78] are commonly used. Consequently, the predicted total energy per particle values have slight variations of the order of 2.6% [81–85].

In the upper panel of Fig. 8, we display the evolution of the ground state energy estimates with mixed and thermodynamic estimators as a function of the imaginary projection time. Although both estimates are statistically indistinguishable, those made with the thermodynamic estimator systematically show a lower value. This behavior is related to the fact that, in contrast to the mixed estimator, the thermodynamic estimator suffers from the finite character of the imaginary time steps since it depends on derivatives of the density matrix. Nonetheless, a strict agreement between different estimators is only expected for $\delta\tau \to 0$. Results obtained with both estimators can be combined, reducing the statistical uncertainty of a final estimate.

Estimates of the potential energy can easily be made at the central bead of the necklace following Eq. (72). As a result, we find $V = -21.61 \pm 0.01$ K. The kinetic energy can then be obtained by subtracting the potential energy from the total energy. For this purpose, we can use either the thermodynamic or the mixed estimator. We found $K_4 = 14.31 \pm 0.01$ K, which is in excellent agreement with experimental data [80, 88, 89] and with theory [30, 90]. We can also calculate the kinetic energy through the thermodynamic estimator of Eq. (74), $K_4 = 14.2 \pm 0.1$ K, a result in agreement with the previous estimate within the statistical errors.

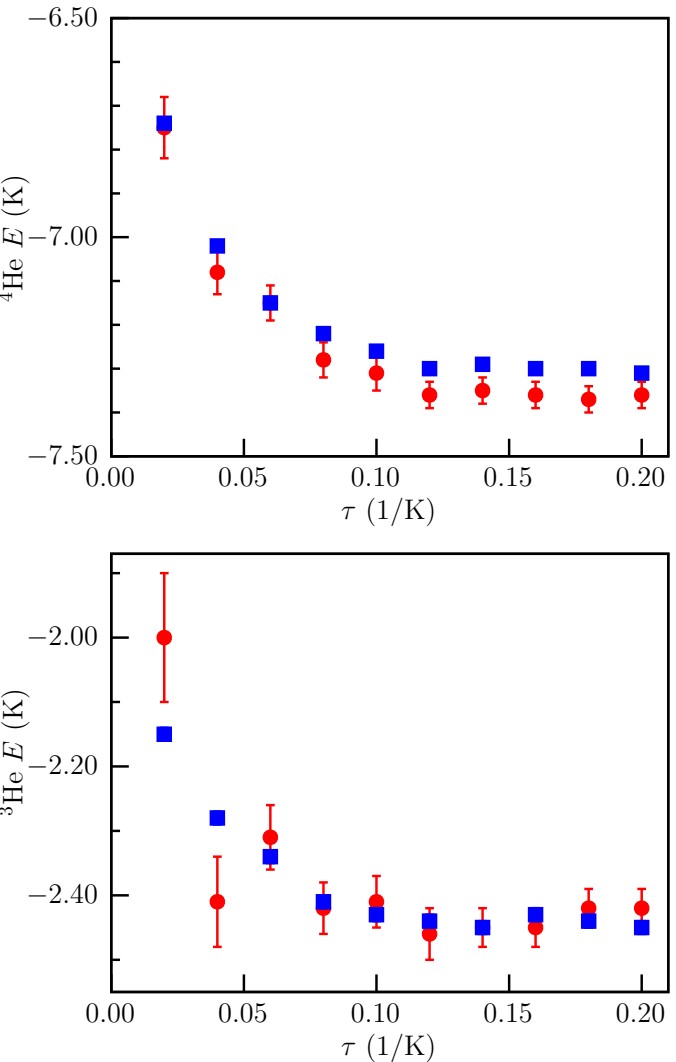

Figure 8: Ground state total energy per particle in units of K as a function of the projection time. The upper panel show results for 108 $^4$He atoms and the lower one for 54 $^3$He atoms. Circles display estimates with the thermodynamic estimator, and squares indicate estimates with the mixed estimator. Statistical uncertainties, when not visible, are smaller than the symbol size.

Table 1: Ground state total energy per particle of each pure system (first column) and densities in units of $\text{\AA}^{-3}$ (second column). Results were obtained using the thermodynamic $E_{\text{th}}$ (third column) and the mixed $E_{\text{mix}}$ estimators (fourth column). In the last columns, the number of particles $N$ considered at each simulation is reported along with the experimental value. All the energies are in units of K.

|        | $\rho$  | $E_{\text{th}}$     | $E_{\text{mix}}$    | $N$  | Experiment        |
|--------|---------|---------------------|---------------------|------|-------------------|
| $^4$He | 0.02186 | $-7.36 \pm 0.03$    | $-7.31 \pm 0.01$    | 108  | $-7.17$[1]        |
| $^3$He | 0.01635 | $-2.44 \pm 0.03$    | $-2.44 \pm 0.02$    | 54   |                   |
| $^3$He | 0.01635 | $-2.37 \pm 0.03$    | $-2.36 \pm 0.02$    | 66   | $-2.47 \pm 0.01$[2] |
| $^3$He | 0.01635 | $-2.34 \pm 0.02$    | $-2.35 \pm 0.01$    | 114  |                   |

[1] Reference [86]

[2] Reference [87]

Table 2: Relative fraction $x$ of $^3$He in $^4$He, at the given densities $\rho$, along with estimated values for the kinetic energy per atom of $^3$He, $K_3$, and $^4$He, $K_4$ at each concentration. Energies are shown in units of K; numerical densities in Å$^{-3}$. Experimental data for pure $^4$He, for a concentration $x = 0.35$ of $^3$He atoms in the mixture and for pure $^3$He are, in this order, from Refs. [88, 93, 94] and obtained at 0.045, 1.96, and 0.5 K, respectively. Simulations for concentration $x = 1$ were made with 114 bodies; in all others the total number of atoms was 108.

| $x$ | $\rho$ | $K_3$ | $K_4$ |
|---|---|---|---|
| 0 | 0.02186 | | $14.31 \pm 0.01$ |
| 0.02 | 0.02175 | $17 \pm 1$ | $14.3 \pm 0.1$ |
| 0.13 | 0.02116 | $16.5 \pm 0.4$ | $13.3 \pm 0.2$ |
| 0.35 | 0.01995 | $15.5 \pm 0.1$ | $12.0 \pm 0.1$ |
| 0.50 | 0.01916 | $14.6 \pm 0.2$ | $11.5 \pm 0.2$ |
| 0.61 | 0.01857 | $14.1 \pm 0.2$ | $10.8 \pm 0.2$ |
| 1 | 0.01635 | $12.34 \pm 0.02$ | |
| Experimental | | | |
| 0 | 0.0218 | | $14.25 \pm 0.3$ |
| 0.35 | 0.01994 | $10.4 \pm 0.3$ | $12.0 \pm 0.6$ |
| 1 | 0.0163 | $12.5 \pm 1.2$ | |

### 7.2.2 Pure $^3$He

We now focus on a strongly correlated Fermi system formed from $^3$He atoms, which also has a long history in the literature [60, 91]. For the liquid $^3$He system, the trial function at the extremities of the necklace was chosen to be a product of a two-body factor of the Jastrow form $\Psi_J$ by a Slater determinant $\Phi_S$ with explicit back-flow correlations [92]

$$\Psi_T(R) = \Psi_J(R)\Phi_S(R), \tag{78}$$

where

$$\Phi_S(R) = \det[\exp(\mathbf{k}_n \cdot \mathbf{x}_m^\uparrow)]\det[\exp(\mathbf{k}_n \cdot \mathbf{x}_m^\downarrow)], \tag{79}$$

with

$$\mathbf{x}_m = \mathbf{r}_m + \sum_{j \neq m}^{N} \eta(r_{mj})\mathbf{r}_{mj}, \tag{80}$$

and $\eta(r)$ given by

$$\eta(r) = \lambda_B \exp[-(r - s_B)/\omega_B] + \lambda'_B/r^3. \tag{81}$$

Here $\lambda_B$, $s_B$, $\omega_B$, and $\lambda'_B$ are variational parameters. The $\uparrow$ ($\downarrow$) symbol corresponds to the spin up (down) configuration.

To study finite size effects in our simulations, we have considered pure unpolarized systems of 54, 66, and 114 $^3$He atoms (corresponding to closed Fermi shells) with the fixed-node approximation and the image action correction. Results for the total energy per particle of the system are presented in Table 1. At density $\rho = 0.01635$ Å$^{-3}$, the results obtained with the thermodynamic estimator applied to the central bead are in excellent agreement with the mixed ones. For the $N = 114$ case, we obtained $E_{th} = -2.34 \pm 0.02$ K and $E_{mix} = -2.35 \pm 0.01$ K. These results agree, within statistical uncertainties, with the ones obtained with $N = 66$ particles, suggesting $N = 66$ a suitable number for pure $^3$He simulations. However, these results do

not agree with the experimental value of $-2.47 \pm 0.01$ K [87]. As predicted in Refs. [95, 96], we found finite size effects in the total energy of approximately 0.1 K between the $N = 54$ and $N = 114$ simulations. Nevertheless, the kinetic energy for a simulation with $N = 114$ atoms, displayed in Table 2, $K_3 = 12.34 \pm 0.02$ K, is in excellent agreement with the experimental result from Ref. [94]. The estimated potential energy is $V = -14.68 \pm 0.01$ K. Finally, the lower panel of Fig. 8 shows the evolution of the binding energy per $^3$He atom as a function of the projection time for the $N = 54$ simulation. As expected, fluctuations in the computed values of the total energy of 54 $^3$He atoms are bigger than those observed for the system of 108 $^4$He atoms (upper panel).

Continued efforts spanning the last four decades to measure the kinetic energy of pure $^3$He face several experimental challenges [93, 94, 97–102]. In Fig. 9, our estimate is compared with some of the most recent experimental and theoretical [92, 103–105] values from the literature. The first attempt to investigate a fermionic system using the PIGS method took an approach that resembles a "released-node" simulation, resulting in a lower kinetic energy [11]. Conversely, considering the Fermi statistics through all beads, as in our approach, yields more accurate results.

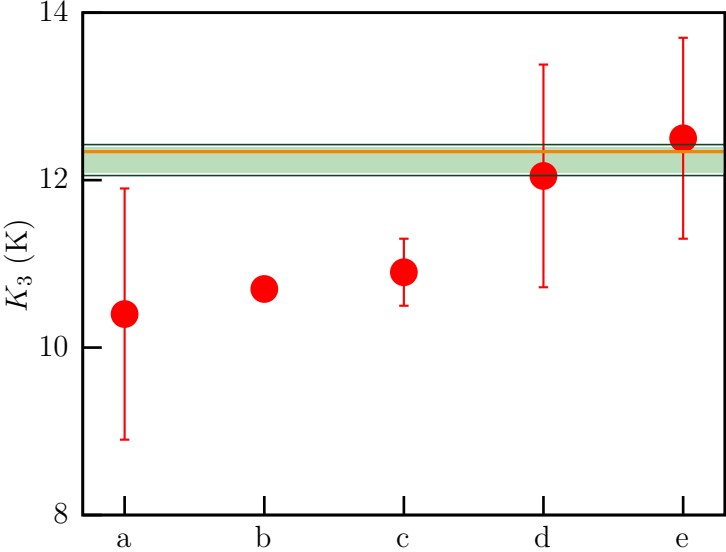

Figure 9: Ground state kinetic energy per atom of liquid $^3$He. The horizontal line within the shaded area corresponds to the FN-PIGS estimate; the statistical error is smaller than the line width. The width of the shaded area represents the span of the theoretical estimates in Refs. [92, 103–105] and their uncertainties. Circles are experimental data from the measured momentum distribution for the references: [a] [100]; [b] [101]; [c] [93]; [d] [102]; [e] [94].

When studying the $^3$He liquid phase, the total pair correlation function and its decomposition into parallel and antiparallel spin components, Eq. (66), is a quantity of interest. In Fig. 10, we displayed our results for the pure system obtained using the converged configurations of the central bead of the necklace. The curve that considers atoms of antiparallel spins has a more pronounced peak since such atoms do not experience the Pauli exclusion principle.

### 7.2.3 The $^3$He-$^4$He mixture

A paradigm of strongly correlated quantum many-body systems composed of both bosons and fermions are mixtures of $^3$He and $^4$He atoms. Despite properties of bulk $^4$He and $^3$He being

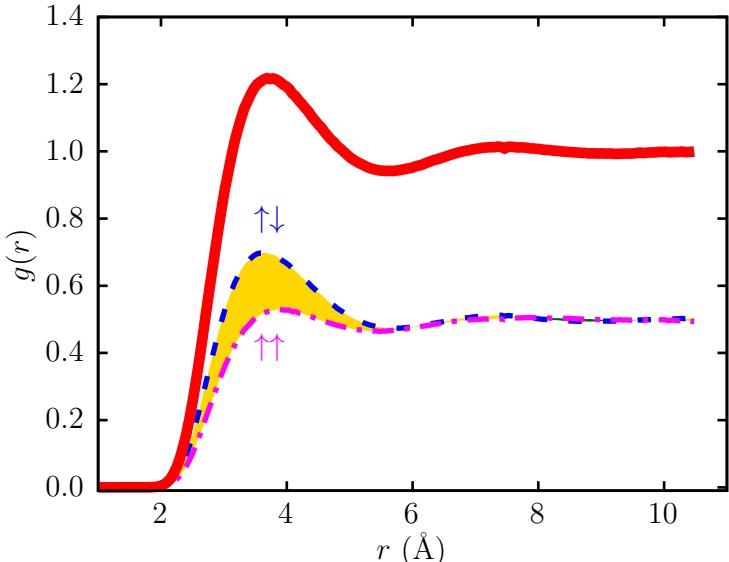

Figure 10: Total pair correlation functions (solid red line) and its components, the antiparallel (dashed blue line) and the parallel spins functions (dashed-dotted magenta line) versus the radial distance. The yellow shaded area shows the region where the pair correlation of antiparallel spin is larger than that of parallel spins. Between $r = 8$ Å and $r = 10$ Å there is a region where the antiparallel pair function is smaller than the one for parallel spins. This region is not easily seen in the figure scale, even though it is depicted in green.

well understood, in the $^3$He-$^4$He mixture, there is still disagreement between experimental data and theoretical results. Such inconsistencies are demonstrated by a recent study of the kinetic energies of each one of the components in the liquid phase at finite temperature [10]. The observed discrepancy for relative concentrations above $x = 0.20$ was partly attributed to the lack of Fermi statistics to describe the $^3$He atoms; they were treated as distinguishable particles. In this context, it is crucial to perform a study where the appropriate quantum statistics treat both components to observe if improvements in the description of the system can be achieved.

With this goal, in simulations using the FN-PIGS method, we describe the liquid helium mixture by a wave function of the Jastrow-Slater form, Eq. (78), which explicitly includes Fermi statistics and back-flow correlations together with the Bose statistics for the $^4$He atoms. However, in the present case, the pair function of the Jastrow form correlates all particles in the mixture $^3$He-$^4$He, with each species and inter-species pair making use of its own variational parameter $b$. Furthermore, the Slater determinant will now also depends on the $^4$He coordinates through the back-flow correlation, increasing the richness of the nodal surface of the system. The parameters we considered for the back-flow correlation were the same as in the description of bulk $^3$He.

We performed simulations with 108 atoms in the mixture and considered different concentrations of $^3$He atoms, $x = N_{^3\text{He}}/(N_{^3\text{He}} + N_{^4\text{He}})$. The equilibrium density of the mixture for each concentration $x$ was calculated using the relation $\rho = [x\, m_3 + (1-x)m_4]/V$, where $V$ is the volume of the simulation cell.

Results for the kinetic energy of both species can be seen in Fig. 11, and the values are displayed in Table 2. The kinetic energies of $^3$He show an improvement in the direction of experimental results, ranging from about 1.5 to 0.4 K from the lower to the higher concentrations, compared with results from the literature [10]. Nevertheless, agreement with experimental

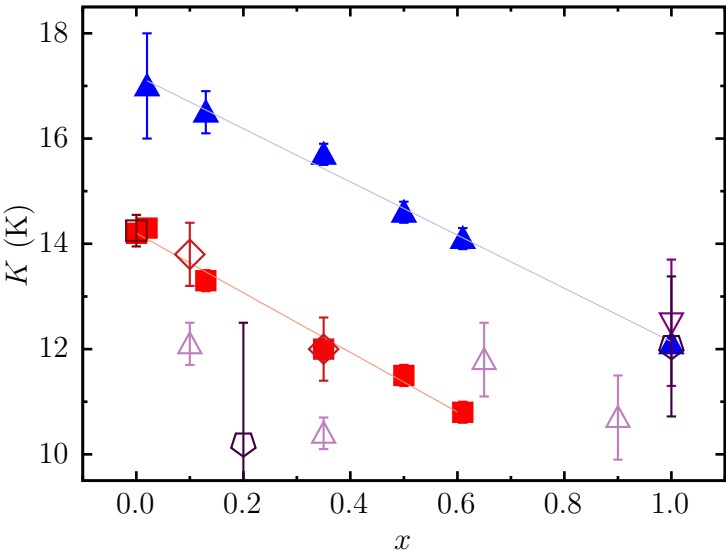

Figure 11: Kinetic energies as a function of $^3$He concentration $x$. The solid squares and triangles represent our $^4$He and $^3$He results, respectively. The empty symbols refer to the most recent experimental data. For $^3$He: empty triangles are from [93]; the upside down triangle is from [94]; the pentagons are from [102]. At $x = 1.0$ (pure $^3$He), our result and an experimental value from [102] are indistinguishable in the figure. For $^4$He: diamonds are from [93]; the empty square is from [88]. The diamond at $x = 0.35$ and the empty square at $x = 0.0$ (pure $^4$He) are indistinguishable from our results; lines are guides to the eye.

data is still lacking because, while these seem to fluctuate around 11 K, theory shows values that decrease with increasing concentrations. However, it is puzzling that, for pure $^3$He, there is an excellent agreement between theoretical and experimental values. Estimated values of the kinetic energy for the $^4$He component not only improve literature results but are also in excellent agreement with experimental data. The result for pure $^4$He corroborates that estimates of kinetic energies made with PIGS are very accurate for bosonic particles.

## 8 Conclusions

We propose an extension of the PIGS method by incorporating the fixed-node approximation, calling it FN-PIGS. As in the bosonic case, FN-PIGS allows estimations without the need for variational results for the extrapolation procedure of any property, regardless of whether it commutes with the Hamiltonian of a fermionic system. Moreover, further developments, such as those that could be achieved through quantum computing, offer a possibility to remove the bias coming from the fixed-node approximation in hybrid quantum-classical algorithms [75], which would make any FN-PIGS estimate numerically exact. It is also possible that other attempts to reduce or even eliminate the sign problem can be easily incorporated into the method [9, 27, 37, 106].

An essential feature of FN-PIGS is that it does not rely on importance sampling transformations [107]. Moreover, the method does not require the introduction of a periodically updated energy shift parameter $E_T$ to stabilize a population of configurations or walkers. This situation introduces bias and is known as the population-control error [108, 109]. Additionally, one can directly distribute parallel processes and periodically collect averages to obtain precise

results with increasingly smaller statistical errors. This approach avoids difficulties associated with strategies where thousands of walkers are submitted to branching algorithms, eventually resulting in the necessity of load balance among processors for efficiency.

We have presented results for a proof-of-concept scenario with the harmonic oscillator to clarify several aspects of implementing the FN-PIGS method. Key features of the method were conveyed by examining properties of mixtures of normal $^3$He in superfluid $^4$He, where both the Fermi-Dirac and the Bose-Einstein statistics intervene, and both species receive a quantum treatment. Our results for different $^3$He concentrations in $^4$He show an improvement in the direction of the experimental values. However, a significant difference between our results and the experimental ones remains for the $^3$He kinetic energies. Indeed, a better knowledge of how the nodal structure of the $^3$He isotope evolves in a mixture with $^4$He is necessary to obtain more accurate results. As put forward in the literature, this disagreement could be related to an underestimation of the $^3$He kinetic energy contribution associated with the tail of the measured momentum distribution. Nonetheless, the kinetic, potential, and total energies obtained with FN-PIGS for pure liquid $^4$He and $^3$He systems at their equilibrium densities agree with the reported experimental and theoretical data. The investigation of the characteristics of fermionic nodes in wave functions is an active research area [110], and hopefully, advances in this direction will contribute to the analysis of $^3$He-$^4$He mixtures.

Our results for pure $^3$He and $^3$He-$^4$He mixtures raise the interesting question of how the dilution of $^3$He in superfluid $^4$He is responsible for shaping the $^3$He nodal structure. After all, results for pure $^3$He and pure $^4$He are more accurate than those obtained for the mixture in the sense that theory and experiment are in excellent agreement. Indeed, the $^3$He-$^4$He mixture is fascinating because, despite being one of the simplest combinations of bosons and fermions, it still offers several challenges for experiments and theory. Results from quantum Monte Carlo methods point to the direction of a $^4$He kinetic energy that decreases with an increasing concentration of $^3$He. Experimental data of $^3$He for this quantity still leave some margin for interpreting its behavior. This scenario makes explicit the need for further studies to understand these systems comprehensively.

FN-PIGS is a robust method that gives accurate results for several physical properties in strongly correlated quantum many-body systems. The method does not rely on variational results to estimate properties like the contact parameter in ultracold quantum cases [111–113], making it a valuable tool in the toolbox of quantum Monte Carlo methods. Potential applications to other systems are not hard to find. For instance, FN-PIGS can be applied in the investigation of neutron matter and ultracold Fermi gases. In nuclear matter at densities lower than $0.003$ fm$^{-3}$, where the complexities of the asymmetric nuclear Hamiltonian can be neglected, a central potential can be used. For quantum Fermi gases, the equation of state, or the spectral weight when an impurity is introduced in a polarized medium, and other properties of interest can be estimated without any extrapolation. For reviews on these topics, see Refs. [114, 115].

As a final remark, in naming the method fixed-node path-integral ground state (FN-PIGS), we follow the standard nomenclature found in the literature when the fixed-node approximation is incorporated into a given existing method. However, we believe that Density Matrix Projection (DMP) better represents how the method operates and reflects its main capabilities. In particular, results can converge to an excited state if, in the extremities of the necklace, the wave functions are orthogonal to the true ground state of the Hamiltonian.

# Acknowledgements

**Funding information**    SAV acknowledges financial support from the Brazilian agency, Fundação de Amparo à Pesquisa do Estado de São Paulo (FAPESP), grant Proc. No. 2016/17612-

7. VZ acknowledges financial support from the Brazilian agencies Coordenação de Aperfenciomante de Pesquisa de Pessoal de Nível Superior under the Netherlands Universities Foundation for International Cooperation exchange program (grant proc. 88887.649143/2021-00) and the Serrapilheira Institute (grant Serra-1812-27802).

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
