# Peer review of "Properties of fermionic systems with the Path-integral ground state method"

_SciPost Physics, doi:SciPost Phys. Core 6, 031 (2023)_

## Round 1 · Referee Report · Anonymous · 2022-10-21

Strengths
This is a well-written and documented contribution that illustrates a slightly different approach to the calculation of ground state expectation values for many-body systems. The proposed methodology is evaluated through a number of relatively simple test cases. This contribution could in principle be useful to other authors interested in experimenting with the approach herein described.
Weaknesses
The main problem that I have with this contribution is that it is really not clear to me in which way the methodology described by the authors can go beyond existing techniques that have been utilized for decades now (mainly Quantum Monte Carlo) to study the same problems treated here by the authors, nor do I see any potential in going beyond anything that has already been done and/or can in principle be straightforwardly done with well-established and well-known methods.
One of the most important limitations of existing quantum Monte Carlo techniques, for example, is the inability to treat systems of fermions exactly, having to resort instead to approximations like the fixed-node, which are largely uncontrolled. The methodology described herein makes use of the same approximation, therefore suffering from the same limitation and it is far from clear to me that it affords any progress even in the (possibly self-consistent) determination, for example, of a more accurate nodal structure.
The results presented by the authors on non-trivial quantum many-body systems are neither original nor (with all due respect) particularly impressive.
Report
While a different way of approaching a problem, or slightly different methodology can always be a valuable addition to the relatively limited toolbox of computational physicists, I am afraid I do not see in this contribution anything of sufficient importance or broad interest to recommend its publication in SciPost Physics. I personally think that this article should be published in a more specialized journal (e.g., Phys. Rev. E)
Author: Bruno Abreu on 2022-10-23 [id 2947]
(in reply to Report 1 on 2022-10-21)
We thank the referee for the time and effort in reviewing our manuscript. We plan to modify the text to highlight aspects of our work that clarify the proposed method's novelty and advantages. However, we directly address a few points below.
i) "... it is really not clear to me in which way the methodology described by the authors can go beyond existing techniques that have been utilized for decades now (mainly Quantum Monte Carlo) ..."
When applied to bosons (Path-Integral Ground State - PIGS), a similar methodology has been employed to simulate various many-body systems (see our Refs.~32-36). The PIGS method complements the finite temperature counterpart, Path Integral Monte Carlo (PIMC), and has several advantages compared to other quantum Monte Carlo methods:
a) it provides unbiased estimates of any quantity without the need for elaborate techniques such as forward walking in Green-Function Monte Carlo (GFMC);
b) it does not approximate the wave function by a finite sum of weighted delta functions among walkers as in Diffusion Monte Carlo (DMC) and GFMC;
c) it does not rely on extrapolations that strongly depend on the wave function accuracy, as in DMC;
d) it does not suffer from fluctuations in the total number of walkers in simulations;
e) it does not require importance sampling.
We stress that our contribution was to expand PIGS to fermionic systems by cleverly considering the anti-symmetrization of the density matrix and the application of the fixed-node (FN) approximation. We prefer to call this extension the Density Matrix Projection method (DMP). In FN-DMC and FN-GFMC, other sources of bias are present in the computation of expected values of operators that do not commute with the Hamiltonian, such as kinetic energies. At finite temperatures, PIMC also shares the advantages of DMP, but dealing with the sign problem in PIMC is much more difficult. As PIGS has done for bosons, our contribution paves the way for a deeper understanding of fermionic systems.
ii) "One of the most important limitations of existing quantum Monte Carlo techniques, for example, is the inability to treat systems of fermions exactly, having to resort instead to approximations like the fixed-node, which are largely uncontrolled."
We very much agree with this limitation. It is present in all Quantum Monte Carlo methods. In fact, a large portion of our effort was towards removing the need for as many approximations as possible. We accomplished to rely solely on the fixed-node approximation to obtain results for mixtures of bosons and fermions with a fully quantum treatment. In our opinion, this is a remarkable result.
iii) "... it is far from clear to me that it affords any progress even in the (possibly self-consistent) determination, for example, of a more accurate nodal structure."
We want to point out that, despite not directly addressing the complex task of determining accurate nodal structures for many-body wavefunctions, another significant contribution of our work is to raise attention to the need for a deeper understanding of how the structure of the fermionic component is affected by the presence of bosons.
iv) "The results presented by the authors on non-trivial quantum many-body systems are neither original nor (with all due respect) particularly impressive."
We want to stress that our results are original and innovative. Unlike DMC and GFMC, the FN approximation is the only source of bias in our estimates of the kinetic energy of liquid He-3. The fact that our energies agree reasonably well with GFMC-calculated ones narrows down the possible sources of mismatch between theory and experiments. Furthermore, our mixture simulations deploy full Bose and Fermi statistics, whereas simulations with PIMC treat fermions with Boltzmann statistics (see our Ref.~10). We obtain improved and more accurate results by considering the correct (anti-)symmetries. Additionally, our results highlight that the complex experimental measurements of the kinetic energy of He-3 must be reassessed. The experimental values are now almost twenty years old; during this period, experimental advancements certainly have occurred.
Sincerely,
The authors
Author: Bruno Abreu on 2022-11-20 [id 3053]
(in reply to Report 3 on 2022-10-27)We thank the referee for the valuable comments and remarks we address below. We will include several changes in the new version of the manuscript that will undoubtedly improve our work's overall quality, readability, and correctness. In response to the referee's primary concern, we will follow the standard Quantum Monte Carlo community nomenclature and call the method Fixed-Node Path Integral Ground State (FN-PIGS). However, as we believe Density Matrix Projection (DMP) is a name that describes precisely how the method works and does not suggest a restriction to ground state calculations, we will recommend adopting it in our conclusions.
1) "after Eq. (18): "It is essential to notice that the DMP is not a variational method". I do not understand the meaning of this phrase: what do the author mean by "variational method"? The energies obtained by Eq. (18) are variational in the meaning that they present an upper bound to the ground state energy, or to some excited state energy when nodal restrictions are imposed. Do the authors simply mean that there are no explicit variational parameters involved? (This is only partially true for fixed-node results)." FN-PIGS is not a variational method because it is not guaranteed to yield a vanishing variance for the total energy of the system, even if the trial wave function is precisely the system's ground state wave function. However, zero variances are possible in the particular case of the mixed estimator, which represents a variational calculation. The same does not hold for other estimators (e.g., thermodynamic). We will change the text in the manuscript accordingly to clarify this point.
2) "Eq. (20): The first equation is wrong: The density matrix as introduced before, Eq. (3), is in general nor fully symmetric nor fully antisymmetric. The first equation of Eq. (20) is only true when using the fully (anti)symmetric density matrix. However, this is not done previously, and is also not done later. In order to justify the result, which is correct (that an explicit (anti-)symmetrisation is not needed when using bosonic/fermionic trial states), the argument must be changed (e.g. it can be seen by relabelling the integration variables involved in Eq. (19), I'm not aware of any argument without integrations, Eq. (20) is simply wrong)." We agree that, in the way it was presented, Eq. (20) is misleading. We thank the referee for noticing that, and we will modify the manuscript accordingly. The correct equation is: \begin{align} (\pm 1)^{{P}}\rho(R,\hat{{ P}}R',\delta\tau)\Psi_T(R') & = \rho(R,\hat{{ P}}R',\delta\tau)\Psi_T(\hat{{P}}R') \rightarrow \rho(R,R',\delta\tau)\Psi_T(R') \nonumber. \end{align}
3) "After Eq. (30): "Equation (30) defines ...., as opposed to the more common mathematical approach." I do not aware of more common mathematical approachs, please provide at least some references." Our approach in this section is guided by physical intuition and the Feynman-Kac formula. In contrast, the more common mathematical approaches that we mention rely on the mathematical manipulation of high-order commutators in the Baker-Campbell-Haussdorff expression [1,2,3], or on general classical mechanics methods. One example is the use of van Vleck determinants to find trajectories with high stability against initial conditions [4,5]. We will include this information and references in the manuscript.
4) "paragraph after Eq. (38): The authors seem to argue that a Jastrow function using a product wavefunction based on the solution of the two-body Schrödinger equation "... captures most ground state correlations in various quantum systems accurately.". This statement is in general wrong as the two-body wave function does not capture any collective effects: the ground state wave function of solids is essentially given by the harmonic approximation. The zero point wave functions of phonons cannot be described by a simple product of two-body solutions. The same is true for the phonon/hydrodynamic part of the ground state wave function of quantum liquids." In Helium systems, typical wavefunctions are of the Jastrow form [6]. A parametrized correlation factor is obtained by considering the leading order of the asymptotic behavior at short-range of a two-body Schrödinger equation. Nonetheless, improvements to the leading order are possible [7]. Corrections due to long-range correlations, as suggested by Chester and Reatto [8,9], can be made based on the quantization of the classical sound field and by taking into account the zero-point motion of longitudinal phonons, as the referee opportunely points out. Although this correction improves the accuracy of the static structure factor at small wave vectors, contributions to the energy are small because the interatomic potential is not long-range. In other systems, it may be very well the case where higher-order correlations are important, which must be addressed case-by-case. We will provide several references covering systems that exhibit superfluidity, supersolidity, Bose-Einstein condensation, and others, where a pair-product many-body wavefunction or pair-product many-body density matrix, along with Quantum Monte Carlo, accurately captures the correlations that are necessary for these inherently quantum, collective phenomena to emerge [10,11,12,13,14]. It is imperative to notice that we are using a many-body wavefunction that is further projected via propagation in imaginary time, thus filtering the actual ground state of the system. Furthermore, we notice that, for the density matrix, the pair-product form is only used at short imaginary time steps (high temperatures). We will modify the text to include a more detailed description of the type of systems where a pair-product wave function is well-known to be a good approach, notably when the main concern is the calculation of energies.
5) "After Eq. (41): "The centrifugal term can be incorporated into the kinetic term and solved analytically" If possible, it would helpful to provide the explicit expressions." Each partial wave $\rho^\ell (r,r',\tau)$ in Eq. (41) is the solution to the following Bloch equation: \begin{equation} -\frac{\partial}{\partial t}\rho^\ell (r,r',t) = \left[ -\lambda_{\text{rel}}\frac{d^2}{dr^2} + \lambda_{\text{rel}}\frac{\ell(\ell + 1)}{r^2} + v(r) \right]\rho^\ell (r,r',t), \nonumber \end{equation} with boundary conditions $\rho^\ell (r,r',0) = \delta(r,r')$ and $\rho^\ell(0, r', t) = 0$. For the free particle case, $v(r) = 0$, the solution is the free-particle contribution in Eq. (42), namely \begin{equation} \rho^\ell_0(r,r',\tau) = \frac{4\pi r r'}{(4\pi\lambda_{\text{rel}}\tau)^{3/2}} \exp \left[ -\frac{(r^2 + r'^2))}{4\lambda_\text{rel} \tau}\right] i_\ell \left( \frac{rr'}{2 \lambda_{\text{rel}}\tau}\right). \nonumber \end{equation} We will include this information in the manuscript.
6) "Last paragraph of section 4.3: "...the larger the imaginary time interval, the smaller the number of effectively contributing waves." This is a bit delicate, since it depends on the distances of the particles considered: at large distances higher and higher partial waves are necessary. However, at low temperatures, sampling of larger distances become more rare." This feature generally only depends on the distance between particles if the interactions are long-range and cannot be neglected for distances that are compatible with the size of the simulation cell, precluding periodic boundary conditions. Even in that case, we can employ remedies such as tail corrections and minimal image strategies (which we do for the Helium potential). For fixed $r$ and $r'$, contributions from partial waves with large angular momenta become increasingly irrelevant with increasing $\delta\tau$, which is a direct consequence of the spherical Bessel function weights. Attempts to sample configurations that enclose larger relative distances become less rare at low temperatures since the free particle Gaussian distribution mainly dictates sampling. This distribution broadens as $\delta\tau$ increases. Such subtlety is at the heart of implementing FN-PIGS and other path-integral methods. Even if one can find an incredibly accurate density matrix at low temperatures, one must still employ a substantial number of beads in the simulation. Using few beads results in large displacements in the bisection algorithm, which tend to be rejected by the repulsive interaction part of the density matrix (particles tend to fall too close to others). The ideal average displacement, and therefore the associated value of $\delta\tau$ and the number of beads, is primarily controlled by the density of the system.
7) "4th paragraph after Eq. (52): "... is increased to > 90% to avoid cross-recross error" Is it possible to be a bit more quantitative, why 90% and how the cross-recross error is obtained?" We monitor the number of attempted moves that result in a sign change of the trial wave function and are discarded due to the fixed node approximation. This number, for acceptance ratios above 90%, represents less than 0.4% of the total attempted moves in a simulation. It is even smaller, 0.02%, for acceptance ratios over 99%. In this scenario, the possibility of having a cross-recross error due to a large displacement is improbable. This is reflected in the fact that ground state energies obtained with acceptance ratios above 90% statistically agree. We should comment that simulations with acceptance ratios over 99% are much longer than those with a 90% ratio (10-20 times). We will include this information in the manuscript.
8) "Section 6: "...operators that have representation in coordinate space...": This is a kind of trivial statement: in the context discussed entirely in this paper, the operators always have a representation in coordinate space. Maybe the authors mean that it is diagonal/local? (Still, all basic operators entering the Hamiltonian, momentum, and positions of particles are diagonal in coordinate space...)." We agree that, in the context of Helium systems, most operators do have representation in coordinate space. However, we want to clarify this point by saying that the FN-PIGS method is, in principle, able to estimate observables derived from any operator for which an analytical expression in coordinate representation can be found, regardless of whether the relevant operator is diagonal in that representation or not. By diagonal, we mean that the position operator eigenstates are also eigenstates of that particular operator. Hamiltonians and other important operators, such as the one-body density matrix, are not generally diagonal in coordinate representation. We will be more specific about this statement in the manuscript and refer to the observables as local/non-local instead of diagonal/non-diagonal.
9) "section 6.2.2: I would have expected also a small discussion of the so-called zero variance principle in the context of the mixed estimator of the energy: If the trial wave function is exact, the local energy is independent of the particle positions, so the statistical fluctuations of this estimator will vanish." The mixed estimator yields the exact energy with zero variance if the exact wave function is known. Nevertheless, in cases where analytical integrations of the wave function cannot be performed, it is convenient to perform a standard MC calculation to estimate the properties of interest. Please, also see the answer to question 1, and note that these modifications will be included in our manuscript.
10) "Figure 6: The oscillations of the values in squares (without image-action) urge for some explanation. Is it related to the time step error (crossing/re-crossing error)?" The cross/recross error is absent in the first exited state of the harmonic oscillator case since the trial wave function has only two pockets. The oscillations are likely related to the fact that, when the image-action is not considered, there is a larger probability for the particle to stay close to the node of the trial wave function compared to the case where the image-action is employed. In other words, the density matrix with the image-action represents more accurately the vanishing of probabilities in the vicinity of the nodal surfaces. We will include this information in the manuscript.
11) "section 7.2.1 the two estimators for the energy ${E_{th}}$ and ${E_{mix}}$ are not "in excellent agreement". They are rather 2 sigma off (which is ok). However, the upper panel of Fig. 8 indicates a problem: although all data point with tau>0.1 are each ok within 2 sigma, since they are statistically independent, the two estimators are roughly sqrt(5)*2 sigmas off which significant." Since the mixed estimator does not depend on derivatives of the density matrix, the energies calculated with the thermodynamic estimator suffer from the finite character of the imaginary-time steps. This error can be removed with a time step extrapolation that we do not perform since it would require several long simulations for each value of $\tau$. Nonetheless, we would like to comment that strict agreement between different estimators is only expected for $\delta \tau \to 0$. Additionally, error bars in these calculations can be made arbitrarily small by running the simulation longer. We will certainly discuss this in our results.
12) "Helium 3: the comparison with experiment must be done with care: Finite size effects are very important for fermions due to shell effects. This is a well known issue which prevents a direct quantitative comparison between results of N=54 and experiment, and the authors should at least mention that. Just for concreteness, the the case discussed (He3 zero pressure) one can expect that the error is around 50mK, e.g. from Phys. Rev. B 74, 104510 (2006) (or some references therein). Therefore, the agreement with experiment found is due to error compensation." We thank the referee for calling attention to this point. Ref. [15] suggests that finite-size effects can be significant. Ref. [16] estimates that, without corrections, values of the total energy are overestimated by at least 50 mK, as mentioned by the referee. We initially chose $N = 54$ He3 atoms to have a closed Fermi surface. In our revised manuscript, we will discuss finite-size effects by including results with $N=66$ and $N=114$ He3 atoms.
Sincerely, The authors
References: [1] Y. Kamibayashi and S. Miura, Variational path integral molecular dynamics and hybrid monte carlo algorithms using a fourth order propagator with applications to molecular systems, The Journal of Chemical Physics 145(7), 074114 (2016-08), doi:10.1063/1.4961149. [2] J. E. Cuervo, P.-N. Roy and M. Boninsegni, Path integral ground state with a fourth-order propagator: Application to condensed helium, The Journal of Chemical Physics 122(11), 114504 (2005), doi:10.1063/1.1872775. [3] S. N. Maximoff and G. E. Scuseria, Exchange energy functionals based on the full fourth-order density matrix expansion, The Journal of Chemical Physics 114(24), 10591 (2001-06), doi:10.1063/1.1373432. [4] N. Makri and W. H. Miller, Correct short time propagator for Feynman path integration by power series expansion in δt, Chemical Physics Letters 151(1), 1 (1988), doi: https://doi.org/10.1016/0009-2614(88)80058-7. [5] N. Makri, Improved Feynman propagators on a grid and non-adiabatic corrections within the path integral framework, Chem. Phys. Lett. 193(5), 435 (1992), doi:10.1016/0009-2614(92)85654-s. [6] W. L. McMillan, Ground state of liquid 4He, Phys. Rev. 138(2A), A442 (1965), doi: 10.1103/PhysRev.138.A442 [7] Y. Lutsyshyn, Weakly parametrized Jastrow ansatz for a strongly correlated Bose system, J. Chem. Phys. 146(12), 124102 (2017), doi:10.1063/1.4978707. [8] G. Chester and L. Reatto, The ground state of liquid helium four, Physics Letters 22(3), 276 (1966-08), doi:10.1016/0031-9163(66)90610-x. [9] L. Reatto and G. V. Chester, Phonons and the properties of a Bose system, Physical Review 155(1), 88 (1967-03), doi:10.1103/physrev.155.88. [10] B. R. de Abreu, F. Cinti and T. Macri, Superstripes and quasicrystals in bosonic systems with hard-soft corona interactions, Physical Review B 105(9), 094505 (2022-03), doi: 10.1103/physrevb.105.094505. [11] V. Abraham and N. J. Mayhall, Coupled electron pair-type approximations for tensor product state wave functions, Journal of Chemical Theory and Computation 18(8), 4856 (2022-07), doi:10.1021/acs.jctc.2c00589. [12] M. Rossi, M. Nava, L. Reatto and D. E. Galli, Exact ground state monte carlo method for bosons without importance sampling, The Journal of Chemical Physics 131(15), 154108 (2009), doi:10.1063/1.3247833. [13] A. Sarsa, K. E. Schmidt and W. R. Magro, A path integral ground state method, J. Chem. Phys. 113(4), 1366 (2000), doi:10.1063/1.481926. [14] C. M. Herdman, P.-N. Roy, R. G. Melko and A. D. Maestro, Entanglement area law in superfluid 4He, Nature Physics 13(6), 556 (2017), doi:10.1038/nphys4075. [15] M. Holzmann, B. Bernu and D. M. Ceperley, Many-body wavefunctions for normal liquid 3He, Physical Review B (Condensed Matter and Materials Physics) 74(10), 104510 (2006), doi:10.1103/PhysRevB.74.104510. [16] F. H. Zong, D. M. Ceperley, S. Moroni and S. Fantoni, The polarization energy of normal liquid 3He, Mol. Phys. 101(11), 1705 (2003), doi:10.1080/0026897031000085119.

---

## Round 1 · Referee Report · Anonymous · 2022-10-27

Strengths
1) detailed description of variational path integral Monte Carlo method
2) simple explicit results on harmonic oscillator model
3) very pedagogical
Weaknesses
1) name "Density Matrix Projection method": The authors correctly refer to already existing names for at least very similar methods, e.g. cariational path integral Monte Carlo, path-integral ground state. I do not see any relevant change compared to these methods which really justifies a new name. This is confusing, in particular, since, as far as I understand, the only change compared to the previous method is to introduce a weight function for off-diagonal configurations. However, in the methodological description, as well as in the application, only diagonal expectation values are treated, without any praticularly new methods nor results.
Report
The paper presents a nice pedagogical review of ground state path integral Monte Carlo method. However, it seems to me that the new proposed "Density Matrix Projection method" provides only rather small extensions to previous methods. Seeing a new method's name in the title, I a bit disappointed by not having seen any major methodological development in this paper; the introduction of a new name for an existing method has misslead me. The results are partially new, but fully consistent with old (backflow) VMC/DMC calculations (the authors do not really push the comparison with more recent results).
In general, the paper is well written, and should be published in one of the SciPost journals.
I have listed some more detailed remarks below which the authors may address in some form.
Requested changes
1) after Eq.(18): "It is essential to notice that the DMP is not a variational method". I do not understand the meaning of this phrase: what do the author mean by "variational method"? The energies obtained by Eq.(18) are variational in the meaning that they present an upper bound to the ground state energy, or to some excited state energy when nodal restrictions are imposed. Do the authors simply mean that there are no explicit variational parameters involved? (This is only partially true for fixed-node results).
2) Eq. (20): The first equation is wrong: The density matrix as introduced before, Eq. (3), is in general nor fully symmetric nor fully antisymmetric.
The first equation of Eq.(20) is only true when using the fully (anti)symmetric density matrix. However, this is not done previously, and is also not done later. In order to justify the result, which is correct (that an explicit (anti-)symmetrisation is not needed when using bosonic/fermionic trial states), the argument must be changed (e.g. it can be seen by relabelling the integration variables involved in Eq.(19), I'm not aware
of any argument without integrations, Eq. (20) is simply wrong).
3) After Eq.(30): "Equation (30) defines ...., as opposed to the more common mathematical approach." I do not aware of more common mathematical approachs, please provide at least some references.
4) paragraph after Eq.(38):
The authors seem to argue that a Jastrow function using a product wavefunction based on the solution of the two-body Schrödinger equation "... captures most ground state correlations in various quantum systems accurately.". This statement is in general wrong as the two-body wave function does not capture any collective effects: the ground state wave function of solids is essentially given by the harmonic approximation. The zero point wave functions of phonons cannot be described by a simple product of two-body solutions. The same is true for the phonon/hydrodynamic part of the ground state wave function of quantum liquids.
5) After Eq.(41): "The centrifugal term can be incorporated into the kinetic term and solved analytically" If possible, it would helpful to provide the explicit expressions.
6) Last paragraph of section 4.3:
"...the larger the imaginary time interval, the smaller the number of effectively contributing waves." This is a bit delicate, since it depends on the distances of the particles considered: at large distances higher and higher partial waves are necessary. However, at low temperatures, sampling of larger distances become more rare.
7) 4th paragraph after Eq.(52): "... is increased to > 90% to avoid cross-recross error" Is it possible to be a bit more quantitative, why 90% and how the cross-recross error is obtained?
8) Section 6: "...operators that have representation in coordinate space...":
This is a kind of trivial statement: in the context discussed entirely in this paper, the operators always have a representation in coordinate space. Maybe the authors mean that it is diagonal/local? (Still, all basic operators entering the Hamiltonian, momentum, and positions of particles are diagonal in coordinate space...).
9) section 6.2.2:
I would have expected also a small discussion of the so-called zero variance principle in the context of the mixed estimator of the energy:
If the trial wave function is exact, the local energy is independent of the particle positions, so the statistical fluctuations of this estimator will vanish.
10) Figure 6: The oscillations of the values in squares (without image-action) urge for some explanation. Is it related to the time step error (crossing/re-crossing error)?
11) section 7.2.1
the two estimators for the energy E_th and E_mix are not "in excellent agreement". They are rather 2 sigma off (which is ok).
However, the upper panel of Fig. 8 indicates a problem: although all data point with tau>0.1 are each ok within 2 sigma, since they are statistically independent, the two estimators are roughly sqrt(5)*2 sigmas off which significant.
12) Helium 3: the comparison with experiment must be done with care:
Finite size effects are very important for fermions due to shell effects.
This is a well known issue which prevents a direct quantitative comparison between results of N=54 and experiment, and the authors should at least mention that.
Just for concreteness, the the case discussed (He3 zero pressure) one can expect that the error is around 50mK, e.g. from Phys. Rev. B 74, 104510 (2006) (or some references therein).
Therefore, the agreement with experiment found is due to error compensation.
Strengths
1.- The details of the method (density matrix projection) are presented excellently.
2.- The method is then applied to bosons, fermions, and mixtures, and it is shown to be robust, at least for the cases studied.
Weaknesses
No obvious weaknesses.
Report
This is an interesting paper. The most attractive aspects of it are that a new method is presented in substantial detail, from which a large community of newcomers and graduate students could easily learn. In fact, the level of clarity in the presentation is so good that I imagine other researchers could use this work to draw connections to their own.
As I am not an expert in the area of Helium, I cannot comment much on the physical results, but the method does appear to be robust, as the authors explain, when calculating for bosons, fermions, and mixtures.
Requested changes
Could the authors comment on the applicability and outlook of their method to neutron and nuclear matter calculations?
Author: Bruno Abreu on 2022-11-20 [id 3052]
(in reply to Report 2 on 2022-10-24)
We thank the referee for carefully reviewing our work and for the positive comments. We address the question about applying our method to nuclear systems below.
Over the past decade, Quantum Monte Carlo (QMC) methods, such as Variational Monte Carlo (VMC), Green Function Monte Carlo (GFMC), and Auxiliary Field Diffusion Monte Carlo (AFDMC), along with realistic nuclear interactions and consistent electroweak currents, led to accurate descriptions of strongly-interacting nuclear systems, including heavy nuclei, neutron matter, and asymmetric nuclear matter.
The Density Matrix Projection (DMP) method is part of the QMC toolbox, presenting general applicability to study strongly correlated many-body systems. For instance, a central potential can be used in neutron matter at a density lower than 0.003 fm$^{-3}$, as the complexities of the asymmetric nuclear Hamiltonian do not need to be considered. This system is similar to ultra-cold Fermi gases, where the equation of state and the spectral weight with impurities in a polarized system can be estimated without any extrapolation (in the sense of diffusion Monte Carlo). For reviews, see Refs. [1] and [2]. In this context, we anticipate future studies of different properties of cold atoms and neutron matter at zero temperature.
The GFMC method has been used to predict nuclei's spectra and electroweak processes with $A\leq 12$ (this limitation arises because GFMC complexity scales exponentially with $A$ due to the spin and isospin sums). DMP shares some significant aspects of GFMC, such as the fermion sign problem. However, the fixed node approximation is the only approximation made in DMP. Furthermore, DMP provides unbiased estimates of any quantity without the need for elaborate techniques, such as forward-walking or extrapolations that depend on the accuracy of the wave function, as in GFMC or Diffusion Monte Carlo (DMC). Additionally, it does not approximate the wave function by a finite sum of weighted delta functions among walkers nor suffers from fluctuations in the total number of walkers in simulations. The propagation to the system's ground state can be achieved with somewhat simplified wave functions. These characteristics of the DMP method could facilitate the study of $A > 12$ nuclei.
As the referee opportunely notes, applying the method to nuclear physics is a relevant prospect, and we will include it in our conclusions. Finally, we mention our decision to follow the standard QMC community nomenclature and call the method Fixed-Node Path Integral Ground State (FN-PIGS) instead of the Density Matrix Projection method. This decision will result in several editorial changes in the manuscript.
Sincerely,
The authors
References:
[1] M. Oertel, M. Hempel, T. Klahn and S. Typel, Equations of state for supernovae and compact stars, Rev. Mod. Phys. 89(1), 015007 (2017), doi:10.1103/revmodphys.89.015007.
[2] J. Carlson, S. Gandolfi, F. Pederiva, S. C. Pieper, R. Schiavilla, K. E. Schmidt and R. B. Wiringa, Quantum Monte Carlo methods for nuclear physics, Rev. Mod. Phys. 87(3), 1067 (2015), doi:10.1103/revmodphys.87.1067.

---

## Round 2 · Referee Report · Anonymous (Referee 1) · 2022-12-19

Report

I thank the authors for their thorough, professional and detailed reply, which however does not make me change my initial recommendation, which continues to be that this article should be resubmitted to a more specialized journal, such as Phys. Rev. E.
It is important that it be clear that I am not disputing that this contribution contains valuable material of interest to other specialists in this field, and therefore warrants publication in some venue. The issue is whether the appropriate publication venue is SciPost physics, and I am sorry but I continue to think that the overall character of this work, namely "a clever way of extending the PIGS method to fermions (but it is important to stress that the PIGS method or its Reptation QMC variant has already been applied to fermions)" is that of an incremental contribution, in my view not the kind that should be considered for publication in this journal.
Obviously, this is nothing but my opinion, and as such, it can be overridden by the other reviewers and/or the Editorial College, but I am asked for an opinion and this is what I can provide, subjective as it undoubtedly is.
  • validity: -
  • significance: -
  • originality: -
  • clarity: -
  • formatting: -
  • grammar: -

Author:  Bruno Abreu  on 2022-12-26  [id 3190]

(in reply to Report 1 on 2022-12-19)
Category:
remark
reply to objection
pointer to related literature

We thank the referee for the report. Once more, we justify and contextualize our work with the comments below.

The toolbox of quantum Monte Carlo methods provides many approaches to investigating many-body systems. To mention a few: Green's function Monte Carlo (GFMC), diffusion Monte Carlo (DMC), path-integral Monte Carlo (PIMC) and the worm algorithm, reptation Monte Carlo (RMC), variational path-integral (VPI) a.k.a. path-integral ground state (PIGS), coupled electron-ion Monte Carlo (CEIMC), and many others. These methods do not compete against each other. In fact, they complement each other.

At zero temperature, GFMC and DMC require guiding functions usually obtained by VMC. PIMC gives exact results (in the Monte Carlo sense) at finite temperatures. CEIMC implemented the RMC method [1,2] to investigate, for instance, molecular hydrogen crystals [3]. In calculations of light nuclei performed with PIGS, the fermionic sign problem was avoided by considering s-wave nuclei ($A \le4$). In this case, RMC is not efficient [4] because path calculations require sums of spin and isospin and, therefore, even if only one bead is removed and added, a recalculation of the whole path is needed.

It is in this context that our work is inserted. To the best of our knowledge, an explicit and detailed treatment of a many-body fermionic system with PIGS using the fixed-node approximation has yet to be published. We expanded beyond just applying PIGS to well-understood quantum many-body systems. We chose the $^3$He-$^4$He mixture, a system that still raises several scientific questions. We have shown that, in contrast to several methods mentioned above, FN-PIGS removes all but one source of bias for estimates in fermionic systems. This formidable achievement paves the way for applications to several other physical systems. Moreover, this work allowed us to discuss, in detail, the subtleties of path-integral calculations not easily found in the literature. Our work will also be valuable to practitioners who want to perform simulations using the path-integrals framework.

Despite the referee's opinion, we are confident that our work is a strong candidate for publication in SciPost Physics. We lament to notice that the referee insists the work must be published in a journal that does not belong to the SciPost Foundation.

Sincerely,
The authors

References
[1] M. Morales, R. Clay, C. Pierleoni and D. Ceperley, First principles methods: A perspective
from quantum monte carlo, Entropy 16(1), 287 (2013-12), doi:10.3390/e16010287.
[2] C. Pierleoni and D. M. Ceperley, Computational methods in coupled electron-ion monte
carlo simulations, ChemPhysChem 6(9), 1872 (2005), doi:10.1002/cphc.200400587.
[3] G. Rillo, M. A. Morales, D. M. Ceperley and C. Pierleoni, Coupled electron-ion monte
carlo simulation of hydrogen molecular crystals, The Journal of Chemical Physics
148(10), 102314 (2018-03), doi:10.1063/1.5001387.
[4] R. Chen and K. E. Schmidt, Path-integral quantum monte carlo calculations of light
nuclei, Physical Review C 106(4), 044327 (2022-10), doi:10.1103/physrevc.106.044327.

---

## Round 2 · Referee Report · Anonymous (Referee 3) · 2023-1-2

Report

The authors considerably improved the manuscript. However, I'm not sure if the paper reaches the very high expectations of Scipost Physics paper, publication in SciPost Physics Core is probably more appropriate.

---

## Round 2 · Author Response

This revised manuscript is a result of the significant effort we have put into addressing each of the concerns raised by the three referees. After the first round of reviews, we are confident that our manuscript presents the subject in a clear and informative way that can be valuable for a broad audience. The several changes and additions in response to the reports contributed to improving our work regarding correctness, completeness, broadness, and readability. A list of the most substantial changes is provided below. Several minor editorial improvements were also performed and are not listed.

---

## Round 2 · List of Changes

• Ref. 3 Weakness: In response to the main concern raised by referee 3, we are naming the method as Fixed-Node Path-Integral Ground State. The title of the manuscript is now "Properties of fermionic systems with the Path-integral ground state method". The entire manuscript was updated accordingly. The following paragraph was added to Section 8 (Conclusions): As a final remark, in naming the method fixed-node path-integral ground state (FN-PIGS), we follow the standard nomenclature found in the literature when the fixed-node approximation is incorporated into a given existing method. However, we believe that Density Matrix Projection (DMP) better represents how the method operates and reflects its main capabilities. In particular, results can converge to an excited state if, in the extremities of the necklace, the wave functions are orthogonal to the true ground state of the Hamiltonian.

  • Ref. 1 Weakness: We made several editorial changes to illustrate our contributions better, emphasizing the novelties our work presents. The most obvious modification is in the abstract, but there are several minor additions and changes to the Introduction and the Conclusions. The abstract now reads: We investigate strongly correlated many-body systems composed of bosons and fermions with a fully quantum treatment using the path-integral ground state method, PIGS. To account for the Fermi-Dirac statistics, we implement the fixed-node approximation into PIGS, which we then call FN-PIGS. In great detail, we discuss the pair density matrices we use to construct the full density operator in coordinate representation, a vital ingredient of the method. We consider the harmonic oscillator as a proof-of-concept and, as a platform representing quantum many-body systems, we explore helium atoms. Pure $^4$He systems demonstrate most of the features of the method. Complementarily, for pure $^3$He, the fixed-node approximation resolves the ubiquitous sign problem stemming from anti-symmetric wave functions. Finally, we investigate $^3$He-$^4$He mixtures, demonstrating the method's robustness. One of the main features of FN-PIGS is its ability to estimate any property at temperature $T = 0$ without any additional bias apart from the FN approximation; biases from long simulations are also excluded. In particular, we calculate the correlation function of pairs of equal and opposite spins and precise values of the $^3$He kinetic energy in the mixture.

  • Ref. 3 Requested change 8: The referee pointed out a concern about using the expression "diagonal in coordinate space" not being clear. We changed the nomenclature from "diagonal" and "non-diagonal" operators to "local" and "non-local" operators. Further clarifications were added to Sections 2 and 6 to elucidate what these two types of operators mean and how they are amenable to FN-PIGS.

  • Ref. 3 Requested change 2: The referee observed an imprecision at the end of Section 2, Eq. 20. We corrected the equation and explained that the expression is the same under integration after relabelling indexes.

  • Ref. 3 Requested change 3: The referee requested clarification on what we meant by ``more common mathematical approaches'' in the context of density matrices (Section 4). We included examples of such mathematical approaches and references to them (Refs. 52 through 56).

  • Ref. 3 Requested change 4: The referee noticed an imprecision in our consideration of wave functions constructed from pair products (Section 4.2). We modified the text to be more precise about in what contexts this type of wave function is well-known to be a good approach. We included examples of systems with references. Following the referee's prompt, we discussed possible improvements.

  • Ref. 3 Requested change 5: The referee requested an explicit expression for the Bloch equation for partial waves that compose the relative-coordinates pair density matrix. We included Eqs. 42 and 43, showing the Bloch equation and the solution for the free-particle case.

  • Ref. 3 Requested change 6: The referee observed the behavior of the partial waves composing the relative-coordinate pair density matrix as the imaginary time interval increases (last paragraph of Section 4.3). We included a detailed discussion about this behavior, making it clear that fewer partial waves are needed as $\delta\tau$ increases. We also added a paragraph about another point the referee raised concerning the likelihood of sampling large relative distances. This ultimately led to a comment about a fundamental aspect of path-integral-based simulations.

  • Ref. 3 Requested change 7: The referee requested more information about the cross-recross error and how we monitor and mitigate it in our simulations (final part of Section 5). We modified the text to clarify how the cross-recross error is avoided and monitored. We included a detailed discussion about the figures involved when different acceptance ratios are considered and an observation about the consequences it implies in simulation times.

  • Ref. 3 Requested changes 1 and 9: The referee requested clarification about how PIGS differs from a variational approach. We removed the phrase in Section 2.2 about PIGS not being a variational method, as we believe that was not the appropriate place to discuss it. We included a paragraph discussing a few specific aspects of the mixed estimator (Section 6.2.2), with a discussion about the zero-variance situation.

  • Ref. 3 Requested change 10: The referee requested a comment on the oscillatory behavior of the energies in Figure 6. We added the comment to the text, explaining the behavior is related to inaccuracies in the expression of the density matrix for the excited state when the image action is not considered.

  • Ref. 3 Requested change 11: The referee made an observation about Section 7.2.1 and the graph shown in Figure 8. We included a paragraph in the manuscript describing the observations in Figure 8 and discussing the observed behavior, with an observation about the limit where different estimators are expected to fully converge.

  • Ref. 3 Requested change 12: The referee requested details about finite-size effects in our fermionic simulations. We performed unsolicited simulations with larger systems to quantify the finite-size effects in calculating energies. We included paragraphs in Section 7.2.2 to discuss these effects, a justification for the number of particles we chose, and listed references in agreement with what we found.

  • Unsolicited changes and additions in Section 7.2.3: We made several changes in this section to better present and discuss our results about the mixture. Several details and a few references were added. The legend of Figure 11 was changed to make the plot clear.

  • Ref. 2 Requested change: The referee requested a discussion about potential applications to nuclear physics systems. We included in our Conclusions references and a discussion about systems we believe are potential candidates to be investigated through FN-PIGS.

---

## Editorial Decision

published